# Near-infrared photoactivatable control of Ca²⁺ signaling and optogenetic immunomodulation

**Lian He[1†], Yuanwei Zhang[2†], Guolin Ma[1†], Peng Tan[1†], Zhanjun Li[2], Shengbing Zang[1], Xiang Wu[2], Ji Jing[1], Shaohai Fang[1], Lijuan Zhou[3], Youjun Wang[3], Yun Huang[1], Patrick G Hogan[4], Gang Han[2*], Yubin Zhou[1,5*]**

[1]Institute of Biosciences and Technology, Texas A&M University Health Science Center, Houston, United States; [2]Department of Biochemistry and Molecular Pharmacology, University of Massachusetts Medical School, Worcester, United States; [3]Beijing Key Laboratory of Gene Resource and Molecular Development, College of Life Sciences, Beijing Normal University, Beijing, China; [4]Division of Signaling and Gene Expression, La Jolla Institute for Allergy and Immunology, La Jolla, United States; [5]Department of Medical Physiology, College of Medicine, Texas A&M University Health Science Center, Temple, United States

**\*For correspondence:** Gang. Han@umassmed.edu (GH); yzhou@ibt.tamhsc.edu (YZho)

[†]These authors contributed equally to this work

**Competing interests:** The author declares that no competing interests exist.

**Abstract** The application of current channelrhodopsin-based optogenetic tools is limited by the lack of strict ion selectivity and the inability to extend the spectra sensitivity into the near-infrared (NIR) tissue transmissible range. Here we present an NIR-stimulable optogenetic platform (termed 'Opto-CRAC') that selectively and remotely controls Ca²⁺ oscillations and Ca²⁺-responsive gene expression to regulate the function of non-excitable cells, including T lymphocytes, macrophages and dendritic cells. When coupled to upconversion nanoparticles, the optogenetic operation window is shifted from the visible range to NIR wavelengths to enable wireless photoactivation of Ca²⁺-dependent signaling and optogenetic modulation of immunoinflammatory responses. In a mouse model of melanoma by using ovalbumin as surrogate tumor antigen, Opto-CRAC has been shown to act as a genetically-encoded 'photoactivatable adjuvant' to improve antigen-specific immune responses to specifically destroy tumor cells. Our study represents a solid step forward towards the goal of achieving remote and wireless control of Ca²⁺-modulated activities with tailored function.

## Introduction

Microbial opsin-based optogenetic technologies have been widely adopted to modulate neural activity (*Fenno et al., 2011*), but similar tools tailored for utilization in non-excitable tissues (*e.g.*, the immune and hematopoietic system) are still limited. The application of channelrhodopsin (ChR)-based optogenetic tools is limited by the lack of ion selectivity and the inability to extend the spectral sensitivity into the near-infrared (NIR) range (*Fenno et al., 2011*). Here we present a tissue penetrable near infrared-stimulable optogenetic platform (termed 'Opto-CRAC') that can be used to reversibly photo-manipulate Ca²⁺ influx through one of the most Ca²⁺-selective ion channels, the Ca²⁺ release-activated Ca²⁺ (CRAC) channel, which is abundantly present in most non-excitable cells (*Hogan et al., 2010*; *Prakriya and Lewis, 2015*). Our tool is based on the engineering of light sensitivity into the CRAC channel and its subsequent coupling to lanthanide-doped upconversion nanoparticles (UCNP), the latter of which act as nanotransducers to convert tissue penetrable NIR light into visible light emission (*Shen et al., 2013*; *Chen et al., 2014*). We demonstrate that Opto-CRAC

**eLife digest** Optogenetics is a technique that has been used to study nerve cells for several years. It involves genetically engineering these cells to produce proteins from light-sensitive bacteria, and results in nerve cells that will either send, or stop sending, nerve impulses when they are exposed to a particular color of light. Neuroscientists have learned a lot about brain circuits using the technique, and now researchers in many other fields are giving it a try.

There are, however, several challenges to using optogenetics in other types of cells. Nerve cells create a tiny electrical impulses when they are activated, which helps them quickly transmit messages. But other types of cells use more diverse means to communicate and transmit signals. This means that optogenetics techniques must be adapted. Additionally, many cells are located deep in the body and so getting the light to them can be difficult.

He, Zhang et al. have now developed an optogenetic system (termed "Opto-CRAC") that can control immune cells buried deep in tissue. The action of immune cells can be tuned by controlling the flow of calcium ions through gate-like proteins in their membranes. He, Zhang et al. genetically engineered immune cells so that a calcium gate-controlling protein became light sensitive. When the cells were exposed to a blue light the calcium ion gates opened. When the light was turned off, the gates closed. More intense light caused more calcium to enter into the cells. Further experiments then revealed that exposing these engineered immune cells to blue light in the laboratory could trigger an immune response.

The next obstacle was getting light to immune cells in a live animal. So, He, Zhang et al. used specific nanoparticles that have been shown to help transmit light deep within tissue. In these experiments, mice were injected with the light-sensitive immune cells and the nanoparticles. Then, a near-infrared laser beam that can transmit into the tissues was pointed at the mice. This caused calcium channels to open in the engineered cells deep in the mice. Finally, further experiments were used to show that this light-based stimulation could boost an immune response to aid the killing of cancer cells. Other scientists will likely use the technique to help them study immune, heart, and other types of cells that use calcium to communicate.

tools can be applied to remotely control $Ca^{2+}$ influx and generate repetitive $Ca^{2+}$ oscillations, photo-tune $Ca^{2+}$-dependent gene expression, and modulate a myriad of $Ca^{2+}$-dependent activities in cells of the immune system, including effector T cell activation, macrophage-mediated inflammasome activation, dendritic cells (DC) maturation and antigen presentation. Our study set the stage for achieving the goal of remote optogenetic immunomodulation and spatiotemporal control over cellular immunotherapy in a wireless manner.

## Results

Following antigen presentation, T cell receptor (TCR) engagement triggers a cascade of signaling events in T lymphocytes that elicit the influx of extracellular $Ca^{2+}$ through the CRAC channel, a classic example of store operated $Ca^{2+}$ entry (SOCE) (*Hogan et al., 2010*; *Prakriya and Lewis, 2015*). The molecular choreography of SOCE is mainly coordinated by two proteins that are located in distinct cellular compartments: (i) ORAI1, a four-pass transmembrane protein that constitutes the CRAC channel pore-forming subunit in the plasma membrane (PM); and (ii) the stromal interaction molecule 1 (STIM1), an ER-resident $Ca^{2+}$ sensor protein that is responsible for sensing ER $Ca^{2+}$ depletion and directly gating ORAI1 channels through its cytosolic domain (STIM1-CT). Store depletion induced $Ca^{2+}$ influx through CRAC channels further activates calcineurin, a downstream $Ca^{2+}$-dependent phosphatase that dephosphorylates the master transcriptional regulator NFAT (nuclear factor of activated T cells) and subsequently causes NFAT nuclear translocation (*Müller and Rao, 2010*). In the presence of the co-stimulatory pathway, which activates the activator protein 1 (AP-1), NFAT cooperates with AP-1 to turn on genes (*e.g.*, IL-2 and IFN-γ) that are characteristic of a productive immune response (*Müller and Rao, 2010*). To enable light control over the $Ca^{2+}$/NFAT pathway, we set out to install light sensitivity into STIM1 by fusing a handful of STIM1-CT fragments with the genetically-encoded photoswitch LOV2 (light, oxygen, voltage) domain (residues 404–546) of *Avena*

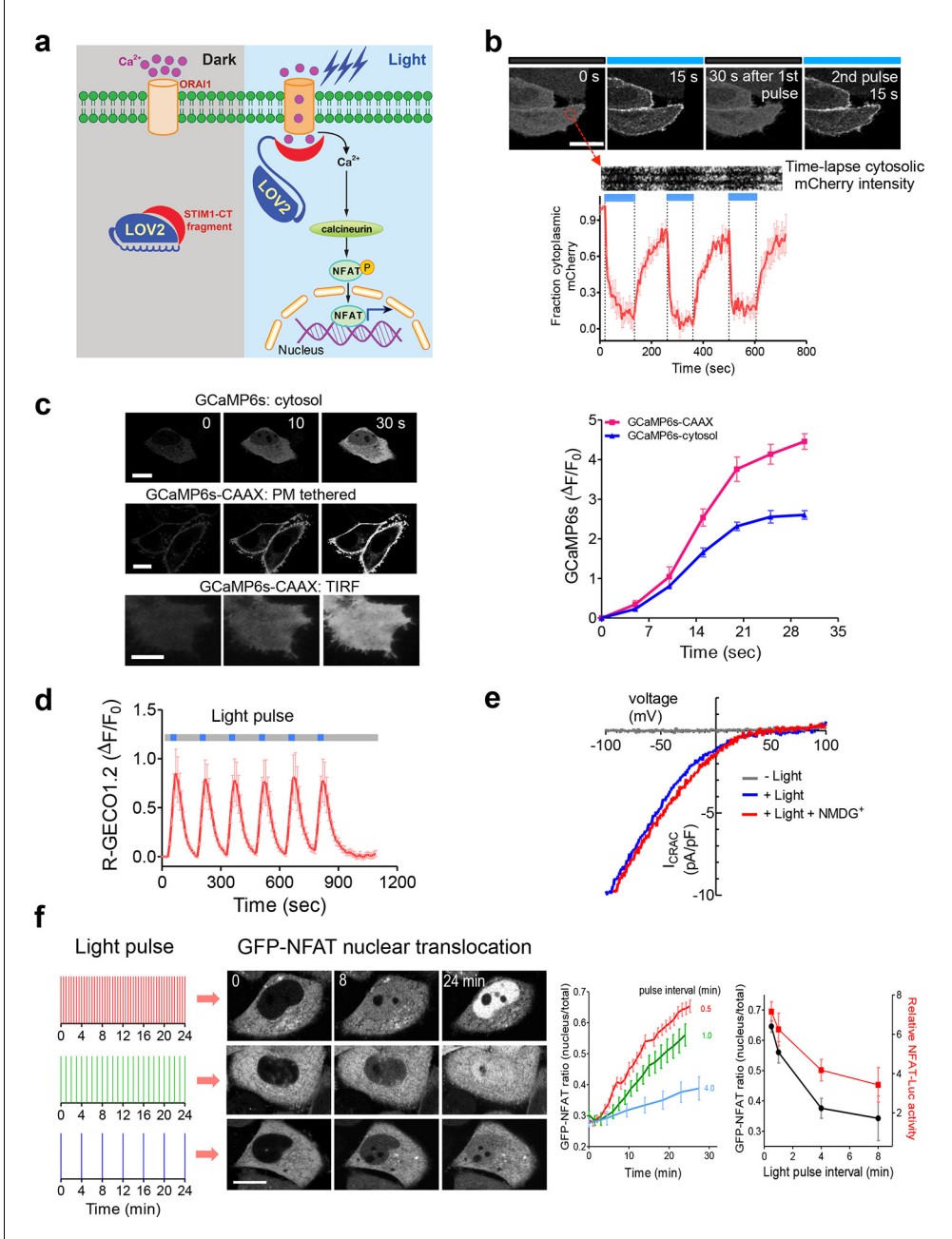

**Figure 1.** LOVSoc-mediated photoactivatable $Ca^{2+}$ entry and nuclear translocation of NFAT in mammalian cells. (**a**), Schematic of light-operated $Ca^{2+}$ entry though engineered Opto-CRAC channels. Fusion with the lightswitch LOV2 domain confers photosensitivity to the ORAI1-activating STIM1-CT fragments. In the dark, STIM1-CT fragments are kept inactive presumably by docking toward the LOV2 domain. Upon blue light illumination, the undocking and unfolding of the LOV2 C-terminal Jα helix lead to the exposure of the STIM1-CT fragments, enabling their interaction with ORAI1 $Ca^{2+}$ channels to trigger $Ca^{2+}$ influx across the plasma membrane. See *Figure 1—figure supplement 1* for the detailed design and comparison among the designed Opto-CRAC constructs. (**b**), Light-inducible translocation of mCherry-$LOV2_{404-546}$-$STIM1_{336-486}$ (designated as mCh-LOVSoc) from the cytosol to the plasma membrane in HEK293T-ORAI1 stable cells. *Upper* panel, the images represent the same cells in the dark (black bar) or exposed to blue light at 470 nm (40 μW/mm²; blue bar). Scale bar, 10 μm. *Lower* panel, Kymograph of mCh-LOVSoc corresponding to the circled area (top) and quantification of mCherry signals over three repeated light-dark cycles (bottom). n = 12 cells from three independent experiments. Error bars denote s.e.m. (**c**), Light-induced $Ca^{2+}$ influx reported by the green genetically-encoded $Ca^{2+}$ indicator (GECI) GCaMP6s. The global cytosolic $Ca^{2+}$ change was monitored after cotransfection of mCh-LOVSoc and GCaMP6s in HeLa cells; whereas the local $Ca^{2+}$ change near the PM was reported by the PM-tethered GCaMP6s-CAAX construct. Shown were representative confocal or TIRF images following blue light stimulation (30 s, 40 μW/mm²). The photo-activated $Ca^{2+}$ response reflected in the fluorescence change was plotted on the right. n = 15 cells from three independent experiments. Error bars denote s.e.m. Scale bar, 10 μm. (**d**), A representative example of light-inducible $Ca^{2+}$ oscillation pattern generated by LOVSoc-expressing HeLa cells

*Figure 1 continued on next page*

*Figure 1 continued*

when exposed to repeated light-dark cycles (30 s ON and 120 s OFF). The red $Ca^{2+}$ sensor, R-GECO1.2, enabled recording of the whole course of intracellular $Ca^{2+}$ fluctuation. n = 8 cells from three independent experiments. Blue bar indicates light stimulation at 470 nm with a power density of 40 $\mu W/mm^2$. Error bars denote s.e.m. (**e**), Photo-triggered current-voltage relationships of CRAC currents in HEK293-ORAI1 cells transfected with mCh-LOVSoc. mCherry positive cells were subjected to whole-cell patch-clamp by a ramp protocol ranging from -100 mV to 100 mV in the presence (blue) or absence (gray) of light illumination. For the red curve, extracellular $Na^+$ was replaced with a non-permeant ion $NMDG^+$ to assess ion selectivity by examining the contribution of $Na^+$. (**f**), Light-tunable nuclear translocation of GFP-NFAT1 and NFAT-dependent luciferase (NFAT-Luc) gene expression in HeLa cells transfected with mCh-LOVSoc. The HeLa-GFP-NFAT1 stable cells were subjected to light pulse stimulation for 30 s whilst the interpulse intervals were varied from 0.5 to 4 min. Representative snapshots of cells during GFP-NFAT1 nuclear translocation were shown in the middle panel. The corresponding time courses and dependence of NFAT nuclear translocation or NFAT-Luc activity on the interpulse interval were plotted on the right. n = 15–20 cells from three independent experiments. Error bars denote s.e.m. Scale bar, 10 μm.

The following figure supplements are available for figure 1:

**Figure supplement 1.** Design and characterization of engineered Opto-CRAC constructs (related to *Figure 1a*).

**Figure supplement 2.** Light-dependent interaction between LOVSoc and ORAI1 (related to *Figure 1b*).

**Figure supplement 3.** Characterization of photoactivatable $Ca^{2+}$ entry into mammalian cells (related to *Figure 1c,d*).

**Figure supplement 4.** Global and local $Ca^{2+}$ influx generated by photo-activation of LOVSoc at defined spatial resolution (related to *Figure 1c*).

**Figure supplement 5.** Schematic representation and light-induced response curves of Opto-CRAC variants reported by GCaMP6s.

**Figure supplement 6.** Examples of light-tunable $Ca^{2+}$ oscillation patterns generated in HeLa cells (related to *Figure 1d*).

*sativa* phototropin 1 (*Christie et al., 1999*; *Harper, 2003*; *Yao et al., 2008*; *Wu et al., 2009*) (*Figure 1a* and *Figure 1—figure supplement 1*). When expressed alone, these STIM1-CT fragments are capable of eliciting varying degrees of constitutive activation of ORAI1 channels to mediate $Ca^{2+}$ entry from the extracellular space to the cytosol (*Yuan et al., 2009*; *Park et al., 2009*; *Zhou et al., 2010a*; *Soboloff et al., 2012*). In the dark, the C-terminal Jα helix docks to the LOV2 domain (*Harper, 2003*; *Yao et al., 2008*; *Wu et al., 2009*) and keeps the ORAI1-activating STIM1-CT fragments quiescent. Upon blue light illumination, photoexcitation generates a covalent adduct between LOV2 residue C450 and the cofactor FMN (*Figure 1—figure supplement 1d*), thereby promoting the undocking and unwinding of the Jα helix to expose the STIM1-CT fragments. Unleashed STIM1-CT fragments further move toward the plasma membrane to directly engage and activate ORAI1 $Ca^{2+}$ channels (*Figure 1a,b*).

We first created a series of Opto-CRAC constructs by varying the length of STIM1-CT fragments, introducing mutations into the LOV2 domain and optimizing the linker between these two moieties (*Figure 1—figure supplement 1a*). After an initial screen of approximately 100 constructs using NFAT nuclear translocation and $Ca^{2+}$ influx as readouts, we decided to use the LOV2-STIM1$_{336-486}$ chimera (designated as 'LOVSoc') in our following experiments because it showed no discernible dark activity and exhibited the highest dynamic range in terms of evoking light-inducible $Ca^{2+}$ influx (*Figure 1—figure supplement 1a,b*). When expressed as an mCherry-tagged fusion protein in HEK293-ORAI1 stable cells, LOVSoc underwent rapid translocation between the cytosol and the PM in response to blue light illumination ($t_{1/2,on}$ = 6.8 ± 2.3 s; $t_{1/2,off}$ = 28.7 ± 6.5 s; *Figure 1b* and *Video 1*). This process could be readily reversed by switching the light off, and could be repeated multiple times without significant loss in the magnitude of response. The light-dependent association between LOVSoc and ORAI1 or ORAI1 C-terminus (ORAI1-CT) was further confirmed by a pulldown assay using purified recombinant proteins and by coimmunoprecipitation assays (*Figure 1—figure supplement 2*). In mammalian cells expressing LOVSoc, the degree of $Ca^{2+}$ influx could be tuned by varying the light power densities (*Figure 1—figure supplement 3a*). After photostimulation for 1 min with a power density of 40 $\mu W/mm^2$ at 470 nm, LOVSoc triggered significant yet varied elevation of cytosolic $Ca^{2+}$ concentrations to approximately 500–800 nM in a dozen of mammalian cell types derived from various non-excitable tissues (*Figure 1—figure supplement 3b*), likely owing to the varied endogenous levels of ORAI proteins among the tested cells. A Light-triggered global

$Ca^{2+}$ influx and oscillations in HeLa or HEK293T cells expressing mCherry-LOVSoc could be monitored in real-time by either Fura-2 (*Figure 1—figure supplement 3c*) or genetically-encoded $Ca^{2+}$ indicators (GECIs), including GCaMP6 (*Figure 1c* and *Videos 2,3*) (*Chen et al., 2013*), R-CaMP2 (*Figure 1—figure supplement 3d*) (*Inoue et al., 2015*), and R-GECO1.2 (*Figure 1d* and *Figure 1—figure supplement 3e*) (*Wu et al., 2013*). Notably, localized light stimulation can be applied to achieve local activation of $Ca^{2+}$ influx at a defined spatial resolution (*Figure 1—figure supplement 4* and *Video 4*), thereby providing a new approach to dissect the effect of $Ca^{2+}$ microdomains in various biological processes (*Parekh, 2008*). Depending on the kinetic properties of the $Ca^{2+}$ indicators used, the half-life time of the cytosolic $Ca^{2+}$ rise in response to light stimulation ranged from 23 s to 36 s. After switching off the light, the cytosolic $Ca^{2+}$ signal decayed with a half-life time of approximately 25–35 s (*Figure 1—figure supplement 3f*). These values are largely in agreement with the time scale of SOCE under physiological stimulation (*Hogan et al., 2010*; *Prakriya and Lewis, 2015*; *Soboloff et al., 2012*). We further measured the photo-activated currents by whole-cell recording in HEK293 cells stably expressing ORAI1 (*Figure 1e*). Following light stimulation, HEK293 cell transfected with LOVSoc developed a typical inward rectifying current, which is characteristic of the CRAC channel and distinct from the greater outward currents of non-selective cation channels such as TRPC (*Prakriya and Lewis, 2015*). Substitution of the most abundant extracellular cation $Na^+$ by a non-permeant ion $NMDG^+$ did not alter the amplitude or overall shape of the CRAC current, implying that $Na^+$ has negligible contribution to LOVSoc- mediated photoactivatable $Ca^{2+}$-selective CRAC currents.

To confer more flexibility to the Opto-CRAC system with varied optical sensitivity, we explored the use of co-expression, membrane tethering or fusion strategies to generate five more variants of Opto-CRAC (*Figure 1—figure supplement 5*). We used either an internal ribosome entry site (IRES)-based bicistronic vector or a self-cleaving 2A peptide strategy (*de Felipe et al., 2006*) to enable the coexpression of ORAI1 and LOVSoc in the same cell with a single vector. Compared to LOVSoc alone, both co-expression systems resulted in ~1.4-fold increase in $Ca^{2+}$ response (*Figure 1—figure supplement 5a*). Tethering LOVSoc to the plasma membrane (PM) with an N-terminal PM-targeting sequence derived from the Src kinase Lyn (*Inoue et al., 2005*) (Lyn11-LOVSoc) expedited the photoactivation process by 3.5-fold (*Figure 1—figure supplement 5b*), presumably owing to its increased local concentration and much closer proximity to the ORAI1 channels. By contrast, a concatemeric form of LOVSoc with two copies covalently connected in a single polypeptide or its fusion to ORAI1 substantially slowed down photoactivatable $Ca^{2+}$ influx (*Figure 1—figure supplement 5c*). Collectively, we have created a set of Opto-CRAC constructs that meet the varying needs on sensitivity and photoactivation kinetics (*Figure 1—figure supplement 5d,e*).

We next asked if we could manipulate the light pulse to generate diverse temporal patterns of $Ca^{2+}$ signals to tune the degree of NFAT activation, which would be reflected in the efficiency of

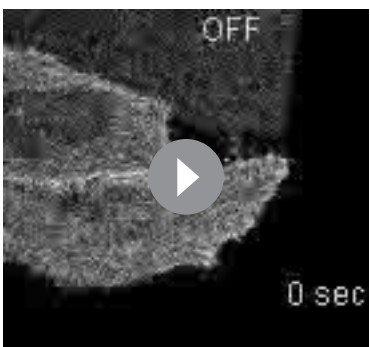

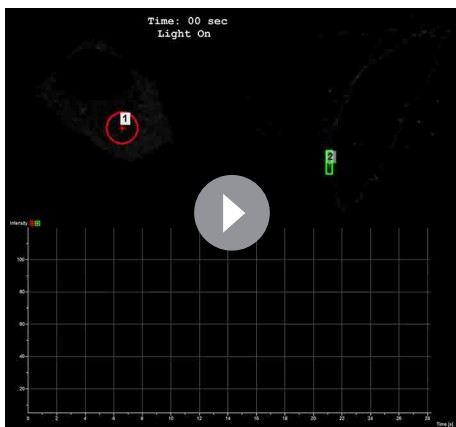

**Video 1.** Light-triggered reversible cytosol-to-PM translocation of mCh-LOVSoc. Three dark-light cycles were applied to HEK293-ORAI1 stable cells transfected with the Opto-CRAC construct mCh-LOVSoc.

**Video 2.** Time-lapse imaging of light-triggered $Ca^{2+}$ influx reported by cytosolic (left) or PM-tethered GCaMP6s. HeLa cells expressing mCh-LOVSoc was exposed to a 488-nm confocal laser.

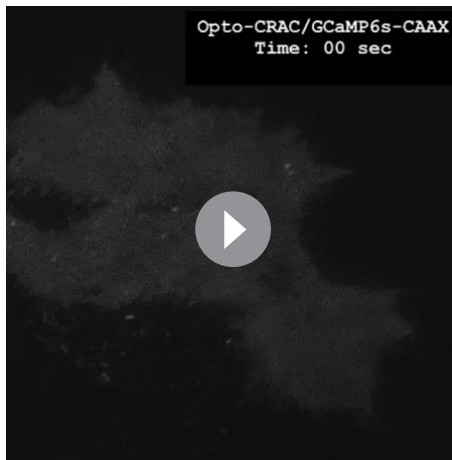

**Video 3.** Light-inducible Ca$^{2+}$ influx monitored with TIRF microscopy in HeLa cells coexpressing GCaMP6s-CAAX and mCh-LOVSoc.

NFAT nuclear translocation and NFAT-dependent luciferase expression. We applied a fixed light pulse of 30 s while varying the interpulse intervals from 0.5 to 4 min to generate Ca$^{2+}$ oscillation patterns with defined temporal resolution (*Figure 1d* and *Figure 1—figure supplement 6*) and compared the levels of NFAT activation in HeLa cells. As shown in *Figure 1f* prolonged interpulse interval was largely accompanied by a decrease in the nuclear accumulation of NFAT. This observation agrees well with previous reports showing that higher Ca$^{2+}$ oscillation frequencies, or faster repetitive Ca$^{2+}$ pulses, tend to increase the ability to activate NFAT (*Lewis et al., 1998*). Thus, we have demonstrated that the engineered Opto-CRAC tools are able to achieve remote and photo-tunable activation of NFAT in mammalian cells (*Figure 1f* and *Video 5*). We further confirmed the NFAT-dependent gene expression in HeLa cells transfected with an NFAT-driven luciferase (NFAT-Luc) reporter construct. In the presence of the co-stimulatory pathway (mimicked by the addition of the pleiotropic PKC activator PMA), light illumination led to a robust increase in luciferase gene expression (*Figure 2a*). A decrease in the light pulse frequency also caused a reduction in the efficiency of Ca$^{2+}$/NFAT-driven luciferase expression (*Figure 1f*). To obviate the use of carcinogenic PMA to photo-trigger gene expression, we also introduced a synthetic 5' transcription regulatory region upstream of gene *Ins1* (*Stanley et al., 2012*), which contains a furin cleavage site that allows

insulin processing in non-beta cells such as HEK293 cells (*Shifrin et al., 2001*). The 5' region is composed of three Ca$^{2+}$-responsive elements in *cis*, including 2–3 copies of serum response elements (SRE), cAMP response elements (CRE) and NFAT response elements with a minimal promoter. Upon light stimulation, we observed a robust production of insulin in cells transfected with LOVSoc, but not in those without LOVSoc expression (*Figure 2a*).

In order to confirm light-inducible gene expression in a more physiologically relevant system, we retrovirally transduced the mCherry-tagged LOVSoc construct into naïve CD4$^+$ T cells isolated from mice (*Figure 2b* and *Figure 2—figure supplement 1*). We then compared the expression levels of two signature genes that are characteristic of activated CD4$^+$ T cells (IL-2 and IFN-γ), in the presence or absence of light illumination, using qRT-PCR and ELISA (*Figure 2b*). Again, in the presence of PMA, light stimulation faithfully mimicked ionomycin-induced effects on the Ca$^{2+}$/NFAT pathway and remarkably boosted the cytokine production by over 15–30 fold in CD4$^+$ T cells transduced with mCh-LOVSoc. By contrast, control cells transduced with the mock retrovirus failed to exhibit light-dependent production of cytokines (*Figure 2—figure supplement 1*). In addition to its well-established role in

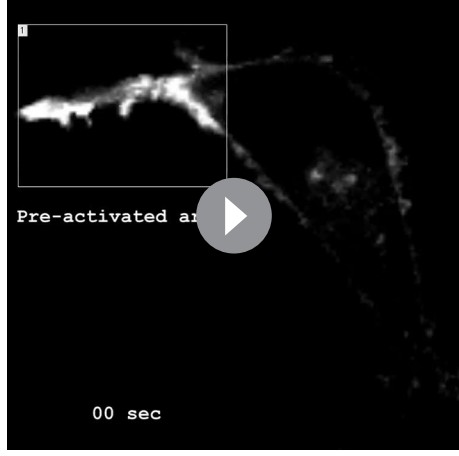

**Video 4.** Sequential and localized activation of Ca$^{2+}$ influx with defined spatial resolution. Imaging was performed on HeLa cells cotranfected with mCh-LOVSoc and GCaMP6s-CAAX. The boxed area was subjected to a brief photostimulation with the 488-nm laser for 10 s, followed by photoexcitation of the whole field at 488 nm to acquire GCaMP6s-CAAX signals. The boxed area showed preactivation of Ca$^{2+}$ influx as reflected by the strong fluorescence signal at time point 0 s; whilst the other areas exhibited a gradual increase in fluorescence intensity following 488-nm light illumination.

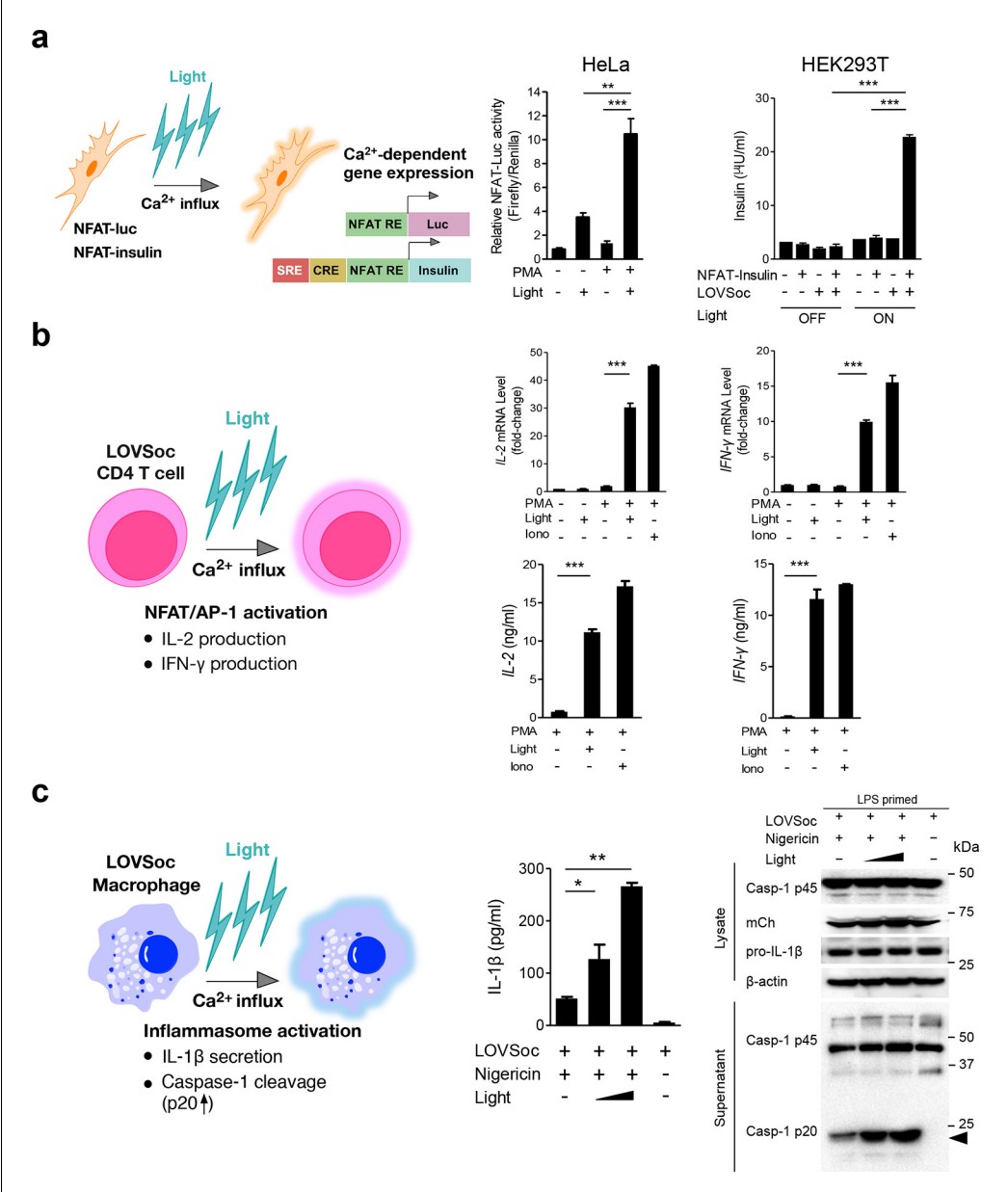

**Figure 2.** Photo-manipulation of $Ca^{2+}$-dependent gene expression and immune response. All data were shown as mean ± s.d. from three independent experiments. *P<0.05; **P<0.01; ***P<0.001 (paired Student's *t*-test). (a), Light-triggered $Ca^{2+}$-dependent gene expression. Cells were either kept in the dark or exposed to pulsed blue light (30 s on with 30 s interval; 40 µW/mm$^2$) for 6 hr prior to cell lysis to quantify luciferase activity (middle) or insulin production (right). Iono, ionomycin. PMA, phorbol 12-myristate 13-acetate. *Left* panel, Schematic of experimental design. Three upstream $Ca^{2+}$-responsive elements in the 5' transcription regulatory region enable efficient initiation of gene expression of the downstream *Ins1* gene encoding insulin following LOVSoc-mediated photoactivatable $Ca^{2+}$ entry and NFAT nuclear translocation. SRE, serum-response element; CRE, cyclic adenosine monophosphate response element; NFAT RE, nuclear factor of activated T cells response element. *Middle* panel, $Ca^{2+}$/NFAT-dependent luciferase activity in HeLa cells transfected with LOVSoc and an NFAT-dependent firefly luciferase reporter vector. A third plasmid encoding the *Renilla* luciferase gene was cotransfected as a reference gene for normalization of gene expression. *Right* panel, Photo-inducible insulin production driven by $Ca^{2+}$-responsive elements in HEK293T cells. (b), Photo-inducible expression of IL-2 and IFN-γ genes in mouse CD4$^+$ T cells expressing the LOVSoc construct. Mouse CD4$^+$ T cells were enriched and purified using an immunomagnetic negative selection kit and transduced with a retrovirus encoding mCh-LOVSoc. On day 5 after transduction and expansion in the presence of IL-2, cells were treated with or without PMA, shielded from light or illuminated with blue light for 8 hr, and then lysed for qPCR (*upper* panels) or ELISA analyses (*lower* panels). The schematic of the experiment was shown on the left. *Upper* panel, Optogenetic stimulation of cytokine production in mouse CD4$^+$ effector T cells transduced with a retrovirus encoding mCh-LOVSoc. *Right* panel, Cytokine production (IL-2 and IFN-γ that are characteristic of activated CD4$^+$ T cells) was determined by ELISA. (c), Photo-tunable amplification of inflammasome activation in macrophages. Human THP-1-derived macrophages were transduced with lentiviruses expressing mCh-

*Figure 2 continued on next page*

*Figure 2 continued*

LOVSoc, primed with LPS (100 ng/ml) and incubated with inflammasome inducer nigericin (10 µM) for 6 hr. Cells were either shielded from light or illuminated with pulsed blue light for 8 hr at power densities of 5 or 40 µW/mm². The cell lysates were collected for ELISA analysis (*left*) and WB (*right*). The schematic of the experiment was shown on the left. *Left* panel, the amount of secreted IL-1β in the culture supernatant quantified by ELISA. *Right* panel, NLRP3 inflammasome activation assessed by Western blotting of lysates and supernatants harvested from cells treated with indicated conditions. Arrowhead, processed caspase 1 (Casp-1) subunit p20.

The following figure supplement is available for figure 2:

**Figure supplement 1.** Retroviral transduction of CD4⁺ T cells and control experiments (related to *Figure 2b*).

driving effector T cell activation, intracellular $Ca^{2+}$ immobilization in macrophage is critical for the activation of the NLRP3 (nucleotide-binding domain, leucine-rich-repeat-containing family, pyrin domain-containing 3) inflammasome (*Murakami et al., 2012*; *Lee et al., 2012*; *Horng, 2014*), which is accompanied by the release of processed caspase-1 (p20 subunit) and the proinflammatory cytokine IL-1β into culture supernatants (*Figure 2c*). Following photostimulation at 5 or 50 µW/mm², we observed a notable light intensity-dependent boost in the production of IL-1β and processed caspase -1 (p20 subunit) in lipopolysaccharide (LPS)-primed THP1-derived macrophages in the presence of a commonly used inflammasome inducer nigericin (*Figure 2c*), thus confirming the feasibility of harnessing the power of light to amplify macrophage-mediated inflammatory responses ex vivo. In aggregate, light-induced activation of the Opto-CRAC channel can generate both global and local $Ca^{2+}$ signals and subsequently cause hallmark physiological responses in both model cellular systems (*e.g.*, HeLa or HEK293 cells) and rodent or human cells of the immune system.

One fundamental roadblock that hampers the application of optogenetic tools in vivo is their inability to stimulate deep within tissues without the use of invasive indwelling fiber optic probes. In order to seek the possibility of controlling the $Ca^{2+}$/NFAT pathway using light in the deep tissue penetrating near-infrared range, we explored the use of lanthanide-doped upconversion nanoparticles (UCNPs) as the NIR light transducer (*Sun et al., 2015*; *Gnach et al., 2015*; *Wu et al., 2015*). Our UCNPs proved to be highly photostable, and their unique upconversion (NIR excitation and emission at visible light range) properties make them an ideal for the remote photoactivation of Opto-CRAC channel activities (*Wu et al., 2009*; *Ostrowski et al., 2012*). In order to match the absorption window of LOV2, we chose mono-dispersed 40-nm β-NaYF₄: Yb, Tm@β-NaYF₄ UCNPs (*Figure 3—figure supplement 1*) that exhibit bright blue emission upon 980 nm CW laser irradiation. When excited at 980 nm, the synthesized UCNPs displayed a sharp emission peak centered around 470 nm (*Figure 3a*). Like direct blue light illumination, UCNPs were able to cause photoactivation of recombinant LOV2 proteins, as reflected by the absorbance changes following NIR light stimulation and the subsequent recovery to the dark state (*Figure 3b*). This finding clearly validates the feasibility of shifting the spectral sensitivity toward the NIR window to activate LOV2-based optogenetic tools.

In order to effectively and specifically illuminate the LOV2-based optogenetic construct in a cellular context, we first developed streptavidin-conjugated UCNPs, then engineered a genetically-encoded streptavidin-binding tag (StrepTag) into the second extracellular loop of the ORAI1 $Ca^{2+}$ channel (mCh-ORAI1^StrepTag, *Figure 3c*) and assessed its capability to recruit streptavidin-

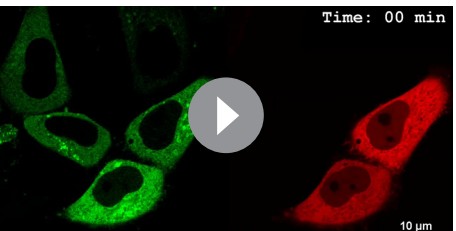

**Video 5.** Light-inducible nuclear translocation of NFAT in HeLa cells. The HeLa GFP-NFAT stable cell line was transiently transfected with mCh-LOVSoc and exposed to pulsed light stimulation at 470 nm (30 s for every 1 min). Shown were fluorescence signals from the green (GFP-NFAT, left panel) and red (mCh-LOVSoc, right panel) channels in the same field. Only cells expressing the Opto-CRAC construct (mCherry-positive, lower right corner) showed light-dependent NFAT nuclear translocation. Note that the cytosol-to-PM translocation of mCh-LOVSoc is not evident as in *Video 1* due to the low expression level of endogenous ORAI1 in HeLa cells and much more abundant expression of mCh-LOVSoc. Nonetheless, the light-triggered activation of endogenous ORAI1 channel was sufficient to activate the downstream GFP-NFAT nuclear translocation.

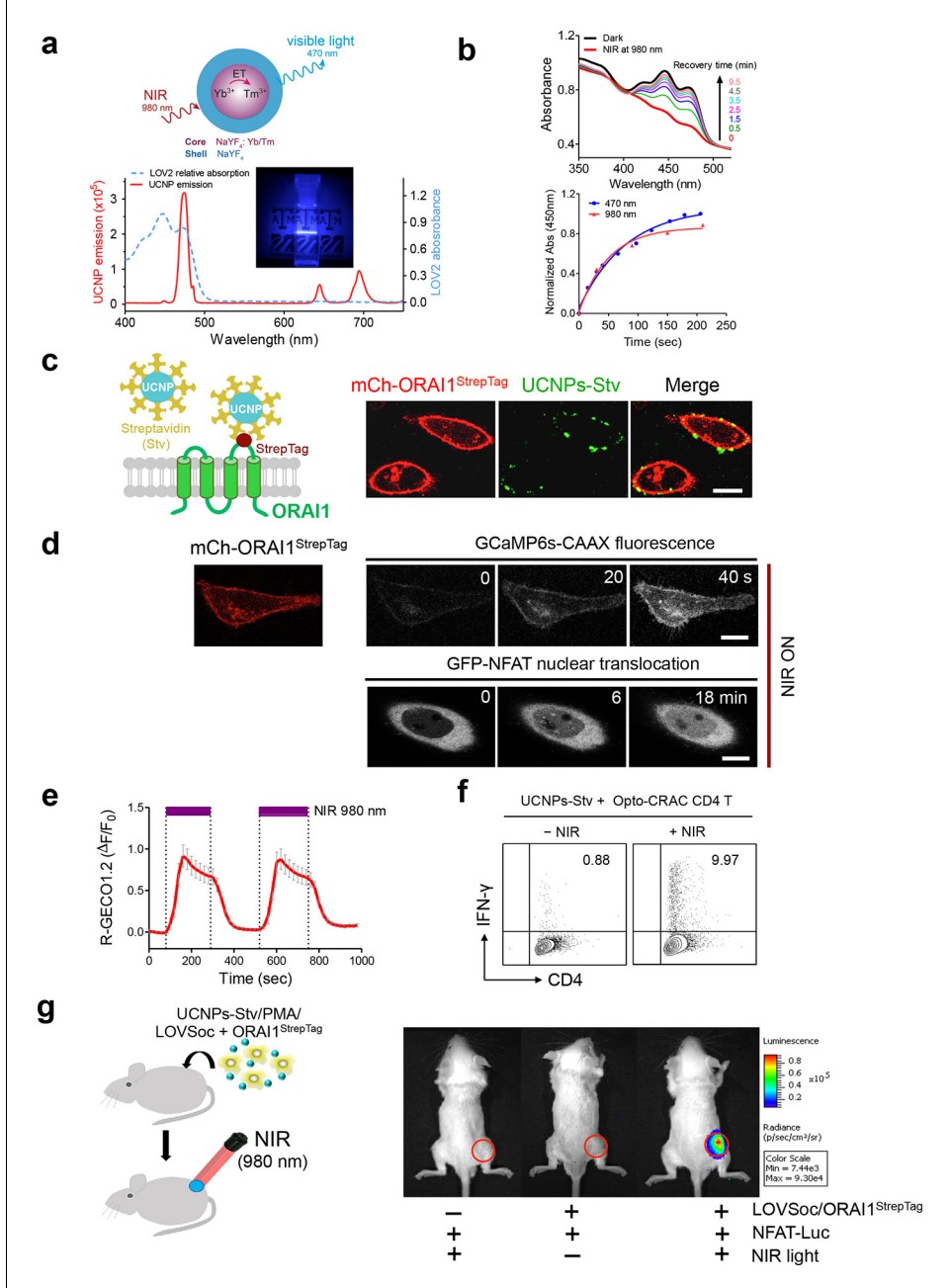

**Figure 3.** NIR light control of Opto-CRAC by lanthanide-doped upconversion nanoparticles. (**a**), Physiochemical properties of the synthesized upconversion nanoparticles. *Upper* panel, schematic illustration of the core/shell structure and energy transfer (ET) among lanthanide ions in the $NaYF_4$: Yb, Tm@$NaYF_4$ upconversion nanoparticles (UCNPs). *Lower* panel, the emission spectrum of $NaYF_4$: Yb, Tm@$NaYF_4$ (solid red line) upon 980 nm CW laser irradiation (15 mW/mm$^2$) superimposed by the absorbance spectrum of recombinant LOV2 protein (dashed blue line). *Inset*: the bright blue emission could efficiently lighten the background upon NIR light illumination at 980 nm. (**b**), NIR light-induced changes in the absorption spectra of purified MBP-LOVSoc at different time interval after mixing with UCNPs-Stv and irradiation with a 980 nm laser (1 min at a power density of 30 mW/mm$^2$). After blue (excited at 470 nm as control, blue circle) or NIR (red triangle) light stimulation, the recovery time course of LOV2 absorbance at 450 nm was plotted in the lower panel. (**c**), Specific targeting of streptavidin-conjugated UCNPs to engineered ORAI1 channels in the plasma membrane of HeLa cells. *Left* panel, Schematic showing the interaction between streptavidin-coated upconversion nanoparticles (UCNPs-Stv) and the engineered ORAI1 Ca$^{2+}$ channel that harbors a streptavidin-binding tag (StrepTag) in the second extracellular loop. The mCh-ORAI1$^{StrepTag}$ protein was able to efficiently recruit and anchor UCNPs-Stv to the plasma membrane of transfected

*Figure 3 continued on next page*

*Figure 3 continued*

HeLa cells. *Right* panel, Florescence microscopy imaging showing the accumulation of UCNPs-Stv (green, $\lambda_{ex}$: 980 nm, $\lambda_{em}$: 450–500 nm) on the plasma membrane of cells transfected with mCh-ORAI1-StrepTag. Scale bar, 10 μm. (d), NIR light-triggered $Ca^{2+}$ influx and NFAT nuclear translocation in HeLa cells coexpressing mCh-ORAI1$^{StrepTag}$ and LOVSoc. $Ca^{2+}$ influx was monitored by GCaMP6s fluorescence whilst GFP-NFAT translocation was reported by GFP signals. Transfected cells were mixed with UCNPs-Stv (20 μg/μl) and illuminated by a 980-nm CW laser to trigger the $Ca^{2+}$ influx. The relatively slow onset of $Ca^{2+}$ influx and NFAT nuclear translocation provided us a time window to quickly capture the green signals without noticeably activating LOVSoc during image acquisition at low excitation energy (<1 μW/mm$^2$). Scale bar, 10 μm. (e), NIR light-induced reversible $Ca^{2+}$ influx reported by R-GECO1.2. HeLa cells were transfected with an IRES bicistronic pMIG retroviral construct that enabled coexpression of ORAI1$^{StrepTag}$ and mCh-LOVSoc. Transfected cells were mixed with 5 mg UCNPs-Stv and illuminated by a 980-nm laser at 30 mW/mm$^2$ to trigger the $Ca^{2+}$ influx. Data were shown as mean ± s.e.m. from 12 cells in two independent experiments. (f), Flow cytometry analysis of IFN-γ production in mouse CD4$^+$ T lymphocytes transduced with retroviruses co-expressing mCh-LOVSoc and ORAI1$^{StrepTag}$. Freshly isolated CD4$^+$ T cells were subjected to in vitro differentiation as described in *Figure 2b*, incubated with 20 μg/μl UCNPs-Stv and 1 μM PMA, and exposed to overnight NIR light pulse (ON/OFF interval of 30 s, 980 nm, 30 mW/mm$^2$) prior to analysis. (g), NFAT-dependent luciferase expression in vivo triggered by NIR light stimulation. *Left,* Schematic of experimental setup. HeLa cells were transfected with NFAT-Luc and constructs encoding LOVSoc/ORAI1$^{StrepTag}$. 48 hr post-transfection, cells were treated with 1 μM PMA, incubated with 10 mg UCNPs-Stv (blue sphere) and implanted to the flanks of mice subcutaneously. The implanted areas were then subjected to NIR light irradiation (red) with a 980 nm CW laser (50 mW/mm$^2$, 30 sec ON, 30 sec OFF for a total of 25 min). *Right,* Shown were bioluminescence imaging of three representative BALB/c mice, one implanted with HeLa cells expressing NFAT-Luc only (*left*) and the other two with cells expressing LOVSoc and NFAT-Luc (*middle* and *right*). Mice were subjected to NIR light irradiation (*left* and *right*) with a 980 nm CW laser. The images were acquired 20 min after receiving a single dose of luciferin (100 μL, 15 mg/ml, *s.c.*). Luciferase-catalyzed bioluminescence was visualized as false color with the same rainbow scale bar for all acquired images. Red circle, implanted area.

The following figure supplements are available for figure 3:

**Figure supplement 1.** Synthesis scheme and in vitro characterization of UCNPs.

**Figure supplement 2.** Green-emitting UCNPs did not activate Opto-CRAC channels upon NIR light stimulation (related to *Figure 3d*).

**Figure supplement 3.** No noticeable heat generation during the in vivo experiment.

conjugated UCNPs (UCNPs-Stv, *Figure 3—figure supplement 1*). In HeLa cells expressing mCh-ORAI1-StrepTag, we detected remarkable local accumulation of UCNPs-Stv on the plasma membrane (*Figure 3c*), confirming the cell-specific targeting of functionalized nanoparticles. To examine whether UCNPs-transduced blue light is sufficient to trigger the opening of Opto-CRAC channels, we monitored cytosolic $Ca^{2+}$ changes using GCaMP6s in HeLa cells co-expressing LOVSoc, mCh-ORAI1-StrepTag and GCaMP6s following NIR light stimulation (980 nm). Within 20 s, transfected HeLa cells exhibited a significant increase in GCaMP6s fluorescence, indicating a rapid rise in the intracellular $Ca^{2+}$ concentration that was evoked by NIR light (*Figure 3d*). This was further confirmed by using a red-emitting $Ca^{2+}$ indicator R-GECO1.2, which enabled recording of reversible $Ca^{2+}$ fluctuation cycles and circumvented the complications associated with potential direct activation of LOVSoc by the green light source used to excite GCaMP6 signals (*Figure 3e*). This increase was found to be caused by $Ca^{2+}$ influx through NIR-to-blue activated Opto-CRAC channels because cells incubated with the control NIR-to-green UCNPs ($\beta$-NaYF$_4$: Yb, 2% Er @ $\beta$-NaYF$_4$; emission maxima at 510 nm) did not show discernible changes in the GCaMP6s signal upon stimulation with the same NIR light (*Figure 3—figure supplement 2*). We then employed NIR light to remotely activate the downstream effector NFAT at the cellular level, and observed NFAT nuclear translocation (*Figure 3d*), as well as NFAT-dependent IFN-γ production in CD4$^+$ T lymphocytes (*Figure 3f*). Next, we sought to demonstrate the potential application of NIR-triggered activation of the Opto-CRAC system in vivo. We performed a proof-of-principle experiment by implanting NFAT-Luc/LOVSoc expressing HeLa cells pre-incubated with UCNPs-Stv subcutaneously in the flanks of mice. The implanted site was irradiated by a 980-nm CW laser outside the body (*Figure 3e*) without noticeable

heat production (*Figure 3—figure supplement 3a,b*) or severe damage to local tissues (*Figure 3—figure supplement 3c*). Luciferase-catalyzed bioluminescence was readily detected after NIR irradiation, whereas no discernible background activation was observed in the negative controls where LOVSoc expression and/or NIR light were absent (*Figure 3g*).

To explore the application of the NIR Opto-CRAC system in a more disease-relevant context, we set out to combine the use of our optogenetic system with DC-mediated immunotherapy in the B16-OVA mouse model of melanoma (*Briles and Kornfeld, 1978*; *Fidler, 1975*), in which ovalbumin (OVA) (*Falo et al., 1995*; *Mayordomo et al., 1995*) is used as a surrogate tumor antigen (*Figure 4a*). Dendritic cells, which provide the essential link between the innate and adaptive immune responses, are adept at capturing tumor antigens and cross-presenting these antigens to T cells in tumor draining lymph nodes (dLNs), thereby sensitizing and generating tumor-specific cytotoxic lymphocytes (CTLs) to cause tumor regression or rejection (*Palucka and Banchereau, 2012*). One of the major challenges of DC vaccination-based immunotherapy is how to efficiently maintain the maturational status of DCs. Pharmacological agents (*e.g.*, ionomycin) or signaling pathways controlling intracellular $Ca^{2+}$ mobilization have been reported to facilitate immature dendritic cell maturation through up-regulation of co-stimulatory molecules CD80 or CD86, major histocompatibility complex (MHC) class I and class II, as well as the chemokine receptor CCR7 (*Félix et al., 2013*; *Matzner et al., 2008*; *Hsu et al., 2001*; *Koski et al., 1999*; *Czerniecki, 1997*). We hypothesize that photoactivatable $Ca^{2+}$ influx in DCs will lead to similar phenotypic changes to expedite and sustain DC maturation and promote antigen presentation, thereby maximally sensitizing T lymphocytes toward tumor antigens to boost anti-tumor immune response. To quickly test this in vitro, we transduced bone marrow-derived DCs (BMDCs) with retroviruses encoding both LOVSoc and ORAI1$^{Strep-Tag}$ (termed 'Opto-CRAC DCs'), pulsed cells with a mixture of OVAp ($_{257}$SIINFEKL$_{264}$) and UCNPs-Stv nanoparticles. NIR light stimulation resulted in approximately 2–8 fold increase in the surface expression of MHC-I/II, CD86, and CCR7 (*Figure 4b*), which are characteristic of matured DCs that are capable of homing to dLNs to interact with T cells to modulate adaptive immune response (*Palucka and Banchereau, 2012*). We next used ex vivo cross-presentation assay to examine how CD8 T cells from OT-1 *Rag1*$^{-/-}$ mice respond to the OVA antigen presented by DCs. The isolated OT-1 CD8 T cells, bearing transgenic T cell receptors that specifically recognize processed OVA peptides (*Clarke et al., 2000*; *Hogquist et al., 1994*), were co-cultured with Opto-CRAC DCs in the presence of OVAp and UCNPs-Stv. After NIR stimulation, co-cultured OT-1 CD8 T cells exhibited over 2-fold increase in both proliferation (*Figure 4c*) and IFN-γ release (*Figure 4d*), clearly attesting to the feasibility of using the NIR-stimulable Opto-CRAC system to expand and photo-prime antigen-specific T cells.

To further validate the immunomodulatory function in vivo, we injected UCNPs-Stv/OVA loaded Opto-CRAC DCs to the B16-OVA murine model of melanoma (*Falo et al., 1995*; *Mayordomo et al., 1995*), in which the B16 tumor cells bearing the OVA antigen could be readily recognized by OT-1 CD8 T cells to elicit anti-tumor immune responses (*Matzner et al., 2008*; *Hsu et al., 2001*). We next adoptively transferred CFSE-abled, OVA-specific OT-I CD8 T cells into the B16-OVA mice and examined their in vivo activation and phenotypic profiles following photoactivatable DC maturation. Compared to the control group shielded from NIR, the proliferation of CD8 T cells was substantially up-regulated after light stimulation, by judging from decreased CFSE staining due to proliferative dilution and increased population of OT-1 CD8T cells in tumor draining LNs and spleens (*Figure 4e*). To assess the functional consequence of immunosensitization of tumor cells toward Opto-CRAC DC-activated immune response, we monitored the tumor growth in mouse melanoma models generated by either subcutaneous or *i.v.* injection of B16-OVA melanoma cells (*Figure 4a*). NIR light stimulation significantly suppressed the tumor growth with diminished tumor volume (*Figure 4f*) or reduced numbers of tumor foci in the lungs (*Figure 4g*). Both our ex vivo and in vivo data converge to support the conclusion that NIR-stimulable Opto-CRAC DC can robustly enhance tumor cell susceptibility to CTL-mediated killing, thereby improving antigen-specific immune responses to selectively destroy tumor cells. By acting as a genetically-encoded 'photoactivatable adjuvant', the Opto-CRAC system may hold high potential for its future use in cancer immunotherapy.

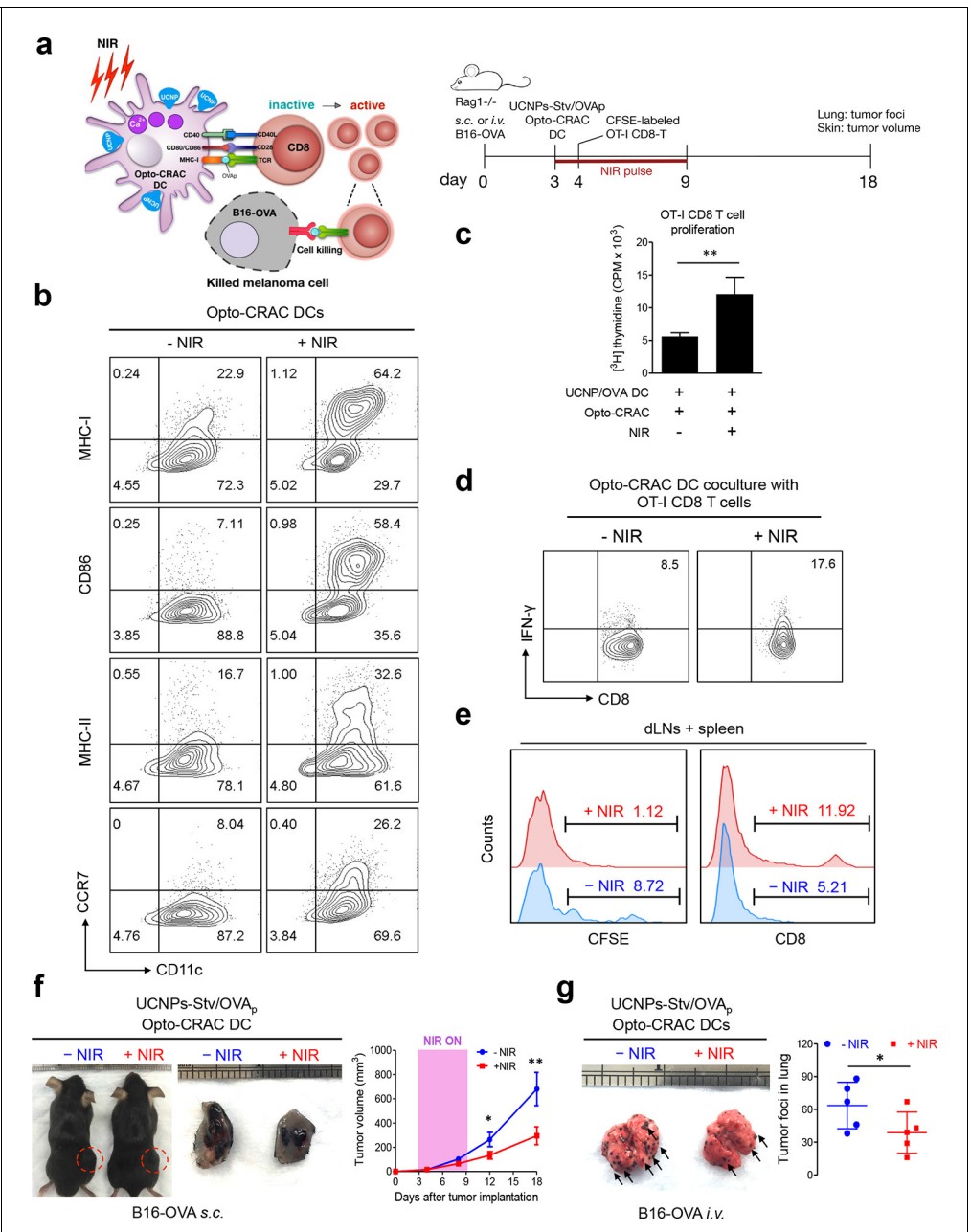

**Figure 4.** NIR light control of Opto-CRAC DC-mediated antigen cross-presentation to OT-I CD8 T cells and B16-OVA melanoma killing. Data were shown as mean ± s.e.m. from at least three independent experiments. *p<0.05; **p<0.01; ***p<0.001 (paired Student's *t*-test). (**a**), Scheme showing the experimental design. NIR-stimulated $Ca^{2+}$ influx in Opto-CRAC DCs prompts immature DC maturation and OVA antigen cross-presentation to activate and boost anti-tumor immune responses mediated by OT-1 CD8 T cells (cytotoxic T lymphocytes, CTLs), thereby sensitizing tumor cells to OVA-specific, CTL-mediated killing in the B16-OVA melanoma model. OVA peptide (OVAp, $_{257}$SIINFEKL$_{264}$) is used here as a surrogate tumor antigen. *Rag1*$^{-/-}$ mice were subcutaneously (*s.c.*) implanted in the flank or intravenously (*i.v.*) injected with 2 x 10$^6$ B16-OVA tumor cells per mice to induce melanoma and lung metastasis. Bone marrow-derived dendritic cells (BMDCs) expressing the Opto-CRAC system (Opto-CRAC DCs) were pulsed with UCNPs-Stv and OVA$_{257-264}$ peptides and injected into *Rag1*$^{-/-}$ mice 3 days after tumor cell injection. Sorted OT-I CD8 cells from OT-I *Rag1*$^{-/-}$ mice, which are labeled by CSFE for monitoring DC antigen cross-presentation and T cell proliferation in vivo, were transferred into B16 tumor-bearing mice one day after Opto-CRAC DC infusion. Mice were kept in dark or exposed to NIR for 1 week (8 hr per day, 1 min ON/OFF pulse, 30 mW/mm$^2$) after DC injection to stimulate Opto-CRAC DC maturation in vivo and photo-boost tumor antigen cross-presentation. 5 days after adoptive transfer, tumor draining lymph nodes (dLNs) and spleen were harvest for FACS (panel **e**) analysis on CFSE-labeled CD8 T cells. Tumor growth was measured by caliper (panel **f**) and mice were sacrificed on day 18 for lung metastasis analysis (panel **g**). (**b**), Flow cytometric analysis on the expression levels of MHC, co-stimulatory and chemokine receptor molecules in BMDCs. Cells were double-immunolabeled with CD11c-FITC *vs* MHC class I-PE, CD86-PE, MHC class II-APC or CCR7-PE and analyzed three days after viral transduction of Opto-CRAC. UCNPs-Stv/ OVA loaded Opto-CRAC DCs were exposed or

*Figure 4 continued on next page*

*Figure 4 continued*

shielded from NIR illumination for 48 hr (30 mW/mm$^2$ with 1 min pulse interval) prior to analysis. (**c**), Proliferation of sorted naïve OT-I CD8 T cells co-cultured with UCNPs-Stv/OVA loaded Opto-CRAC DCs, with or without NIR illumination, was measured by the [$^3$H] thymidine incorporation assay. (**d**), Flow cytometric analysis of IFN-γ production in sorted naïve OT-I CD8 T cells co-cultured with UCNPs-Stv/OVA loaded Opto-CRAC DCs with or without NIR stimulation. (**e**), Flow cytometric analysis of in vivo proliferation of CFSE-APC labeled OT-I CD8 T cells in dLNs and spleen 6 days after injection of UCNPs-Stv/OVA loaded Opto-CRAC DCs with or without NIR pulse excitation (30 mW/mm$^2$) as indicated in panel a. (**f**), Tumor-inoculated sites (*left*) were isolated from tumor-bearing mice (n = 5) shielded or exposed to NIR and the tumor sizes (mm$^3$) were measured at indicated time points shown in the growth curve (*right*) after tumor implantation. (**g**), Representative lungs with melanoma metastases (*left*) were isolated from tumor-bearing mice shielded or exposed to NIR. The histogram represents counted numbers of visible pigmented tumor foci (as exemplified by the arrows) with pulmonary melanoma metastases on the surface of lungs (*right panel;* n = 5 mice).

## Discussion

In the present study, we described an NIR-stimulable optogenetic platform based on engineered CRAC channels and lanthanide-doped upconversion nanoparticles. Depending on the pulse and intensity of light input, the photosensitive module, LOVSoc, can reversibly generate both sustained and oscillatory Ca$^{2+}$ signals. The magnitude and kinetics of photo-activated Ca$^{2+}$ influx largely mimic the physiological responses following engagement of immunoreceptors or ligand binding to its cognate membrane receptors that leads to store depletion (*Prakriya and Lewis, 2015*). Ectopic expression of a single component of LOVSoc at endogenous levels of ORAI is sufficient to elicit strong intracellular Ca$^{2+}$ elevation in a dozen of cell types derived from a wide range of human or rodent tissues. Most critically, light-generated Ca$^{2+}$ signals can further lead to hallmark physiological responses in cells of the immune system. The sensitivity and photoactivation kinetics of this system can be further tuned by tethering LOVSoc to PM or through co-expression and fusion with ORAI1. When paired with deep tissue-penetrant and NIR-stimulable UCNPs, we have successfully demonstrated the potential application of our Opto-CRAC system to drive Ca$^{2+}$-dependent gene expression and to photo-modulate immune response both in vivo and ex vivo. Compared to other existing optical tools, our Opto-CRAC system that has several distinctive features: First, complementary to the existing ChR-based tools that exhibit less stringent ion selectivity and tend to perturb intracellular pH due to high proton permeability, our Opto-CRAC system is engineered from a bona fide Ca$^{2+}$ channel that is regarded as one of the most Ca$^{2+}$-selective ion channels. Although the unitary conductance of CRAC channel is estimated to be low (<10 fS in 2 mM extracellular Ca$^{2+}$ in T cells; compared to 4–10 pS for voltage-gated Ca$^{2+}$ channels) (*Prakriya and Lewis, 2015*; *Zweifach and Lewis, 1993*), sustained Ca$^{2+}$ influx (up to minutes) through native ORAI1 channels is sufficient to activate downstream effectors. The high Ca$^{2+}$ selectivity (P$_{Ca}$/P$_{Na}$: >1000) and its small unitary conductance is speculated to reduce the energy requirement of pumping out Na$^+$ during sustained Ca$^{2+}$ entry, thereby enhancing the specificity of downstream effector function (*Prakriya and Lewis, 2015*). Second, the Opto-CRAC tool has a relatively small size (<900 bp, compared to >2.2 kb of ChR) and is thus compatible with almost all existing viral vectors used for in vivo gene delivery. Indeed, we have successfully used retroviral and lentiviral expression systems to deliver Opto-CRAC into primary T cells, macrophages and dendritic cells. Its potential delivery into excitable tissues (*e. g.*, muscle, heart and brain) using adeno-associated viruses remains to be tested in follow-on studies. Third, the tunable and relatively slow kinetics make it most suitable for interrogating Ca$^{2+}$-modulated functions in non-excitable cell types, such as cells in the endocrine, immune and hematopoietic system. We find that our system may find broad use in adoptive cell transfer experiments or adoptive immunotherapies, which are widely used in both basic research and the clinic settings (*Palucka and Banchereau, 2012*; *Restifo et al., 2012*). Fourth, in conjunction with upconversion nano-transducers, the light harvesting window can be shifted to the NIR region where deep tissue penetration and remote stimulation are feasible. Results from our in vivo studies clearly indicate that the Opto-CRAC channel and its downstream effectors can be remotely activated using NIR light, thereby paving the way for its future applications in more (patho)physiologically-relevant mouse models, or ultimately, in cancer immunotherapies with improved spatiotemporal control over engineered therapeutic T cells or DCs to reduce off-tumor cross-reaction and mitigate toxicity (*Morgan et al., 2010*). Given the spatial and temporal accuracy of NIR light, it is also possible to use guided NIR light to confine localized blue light generation, thus avoiding the photoactivation of off-

target regions. Lastly, but critically, the lanthanide-doped UCNPs can be applied to activate other optogenetic tools that are dependent on blue light-absorbing cofactors (*e.g.*, ChR2 and CRY2). We anticipate that the flexible adaptability of our novel approach will lead to new opportunities to fine-tune $Ca^{2+}$-dependent immune responses and interrogate other light-controllable cellular processes while minimally interfering with the host's physiology.

## Materials and methods

### Chemicals and antibodies

Fura-2 AM calcium indicator was purchased form Life Technologies (Carlsbad, CA, USA). Phorbol 12-myristate 12-acetate (PMA), ionomycin, thapsigargin (TG) and isopropyl-ginethiogalactopyrano-side (IPTG) were purchased from Sigma Aldrich (St Louis, MO, USA). Tri(2-carboxyethyl)phosphine (TCEP) was obtained from Pierce (Life Technologies). Amylose resin used for MBP pulldown was purchased from New England Biolabs (Ipswich, MA, USA). Ni-NTA resin used for purification of GB1-ORAI1-CT was purchased from Qiagen (Valencia, CA, USA). The mouse monoclonal anti-Flag M2-HRP (A859, Sigma-Aldrich, St. Louis, MO, USA) antibody, the rabbit anti-mCherry polyclonal antibody (NBP2-25157, Novus Biologicals, Littleton, CO, USA), the rabbit anti-Caspase-1 antibody (D7F10, Cell signaling, Danvers, MA, USA) and the rabbit anti-IL-1$\beta$ antibody (sc-7884, Santa Cruz Biotechnology, Dallas, TX, USA) were used at a 1:1000 dilution. For flow cytometry (FACS) analysis, anti-mouse MHC Class II (I-A/I-E) APC (), anti-mouse IFN gamma PE (), anti-mouse CD86 (B7-2) PE (12–7311), anti-mouse CD197 (CCR7) PE (12–1971), anti-mouse MHC Class I PE (12–9558), anti-mouse CD11c FITC (11–0114), anti-mouse CD4 PerCP-Cyanine5.5 (45–0042), and anti-mouse CD8a APC (17–0081) were purchased from eBioscience. All other reagents were form Sigma-Aldrich unless otherwise indicated.

### Plasmids

#### Constructs for fluorescence imaging and luciferase assays

The pTriEX-mcherry-PA-Rac1 plasmid was purchased from Addgene (#22027). STIM1-CT fragments (residues 336–450, 336–460, 336–473, 336–486, 342–486, 344–486) were amplified using the KOD hot start DNA polymerase (EMD Millipore, Billerica, MA, USA) and inserted downstream of LOV2$_{404-546}$ between HindIII-XhoI restriction sites to replace Rac1. The LOV2 fused STIM1$_{233-450}$ construct (LOVS1K) (*Pham et al., 2011*) was purchased from Addgene (#31981). The short linker (KL) between LOV2 and STIM1-CT fragments was made by replacing Rac1 with STIM1$_{336-486}$ in the vector pTriEx-mcherry-PA-Rac1 using HindIII-XhoI sites; whilst the NotI-XhoI sites were used for producing a long linker (KLAAA). Mutations in the LOV2 domain were introduced by using the QuikChange Lightning Multi Site-Directed Mutagenesis Kit (Agilent Technologies, Santa Clara, CA, USA) by following the manufacture's protocol. pcDNA3.1-mCherry-ORAI1 were generated by sequential insertion of mCherry in the BamHI-EcoRI sites and human ORAI1 gene between EcoRI and XhoI sites of the vector pCDNA3.1(+) (Life Technologies). Next, oligos encoding the StrepTag (WSHPQFEK) (*Schmidt and Skerra, 2007*) were inserted into the second extracellular loop of ORAI1 after residue 208 through a standard PCR method to construct pcDNA3.1-mCherry-ORAI1-StrepTag. pGP-CMV-GCaMP6s-CAAX (#52228), pGP-CMV-GCaMP6m (#40754) and CMV-R-GECO1.2 (#45494) were obtained from Addgene. The firefly luciferase reporter vector pGL4.30[luc2P/NFAT-RE/Hygro] (abbreviated as NFAT-Luc) and the control *Renilla* luciferase reporter plasmid pRL-TK were purchased from Promega (Madison, WI, USA). The red calcium sensor pN1-R-CaMP2 was a gift from Dr. Haruhiko Bito at University of Tokyo, Japan.

#### Constructs for co-expression of LOVSoc with ORAI1

A murine stem cell virus (MSCV)-based vector pMIG was obtained from addgene (#9044). This bicistronic IRES-GFP containing retroviral was used for insertion of cDNA sequences encoding mCherry-LOV2-STIM1$_{336-486}$ between the XhoI and EcoRI restriction sites. The pMIG-mCh-LOVSoc plasmid, along with the empty vector as control, was used for retroviral transduction of isolated mouse CD4$^+$ T or dendritic cells. In a further modified version, GFP was replaced by cDNAs encoding WT or engineered ORAI1 that contain a StrepTag in its second extracellular loop to recruit UCNPs-Stv, thus allowing bicistronic expression of both LOVSoc and ORAI1 in the same construct. To enable co-

expression at ~1:1 ratio, cDNAs encoding mCh-LOVSOC and ORAI1[StrepTag] were connected by a self-cleaving 2A peptide sequence (*de Felipe et al., 2006*) and inserted into the pTriEx vector for transient expression or into a LeGO lentiviral vector for transduction of human or rodent primary cells.

### Constructs for recombinant protein expression in *E.Coli*

The DNA sequences encoding LOVSoc described above were amplified and inserted into the vector pMCSG9 between the BamHI and XhoI sites for expression as MBP-LOVSoc protein. To construct a bacterial expression plasmid of ORAI1-CT (residues 259–301) fused with the B1 domain of streptococcal protein G (GB1), the GB1 gene was inserted between NcoI-BamHI sites and ORAI1-CT was subsequenlty inserted between the BamHI and XhoI sites of the host vector pProEx-HTb (Life Technologies). The GB1 tag was used as a small tag to enhance the protein solubility and aid affinity purification.

## Fluorescence Imaging and total internal reflection fluorescence (TIRF) microscopy

HeLa, HEK293/HEK293T and other indicated immortalized cell lines from the American Type Culture Collection (ATCC) were cultured in Dulbecco's modified Eagle's medium (DMEM, Sigma-Aldrich) supplemented with 10 mM HEPES and 10% heat-inactivated fetal bovine serum. All the cells were grown at 37°C in a 5% $CO_2$ atmosphere. Cultured cells were seeded on 35-mm glass bottom dishes and an inverted Nikon Eclipse Ti-E microscope customized with Nikon A1R+ confocal laser sources (405/488/561/640 nm) was used for confocal imaging. The same microscope body connected to a Ti-TIRF E motorized illuminator unit (488 nm/20 mW and 561 nm/20 mW lasers) with a 60×, NA 1.49 oil-immersion TIRF objective was used for TIRF imaging. 100-nm fluorescent beads (TetraSpeck microspheres, Life Technologies) were deposited onto a coverslip and imaged as markers for later alignment.

To monitor mCh-LOVSoc translocation from the cytosol to PM, 50–100 ng pTriEx-mCherry-LOV-Soc was transfected to HEK293-ORAI1 stable cells using Lipofectamine 3000 (Life Technologies). Cells were imaged 24 hr after transfection. Photostimulation was provided by an external blue light (470 nm, tunable intensity of 0–50 $\mu W/mm^2$, ThorLabs Inc., Newton, NJ, USA). Light power density was measured by using an optical power meter from ThorLabs. Light cycles were applied either manually or programmed by connecting to a DC2100 LED Driver with pulse modulation (ThorLabs). Time-lapse imaging of mCherry signal was carried out in the dark by turning on only the 561-nm laser channel.

For measurements of $Ca^{2+}$ influx using the green color calcium indicator GCaMP6s, 50–100 ng mCh-LOVSoc and 100 ng cytosolic GCaMP6s or membrane-tethered GCaMP6s-CAAX were cotransfected into HeLa or HEK293T or other indicated cells using Lipofectamine 3000. Twenty-four hours after transfection, a 488-nm laser was used to excite GFP, and a 561-nm laser to excite mCherry at intervals of 1–5 s. The mCherry-positive cells were selected for statistical analysis. Since the excitation wavelength used to acquire the GCaMP6s signals (488 nm) partially overlaps with the photo-activating wavelength of LOVSoc, $Ca^{2+}$ influx was elicited when the 488-nm laser source was turned on, and thus GCaMP6s could only be used to monitor the ON phase of $Ca^{2+}$ flux. For localized photostimulation, we took advantage of the NIKON component designed for fluorescence recovery after photobleaching (FRAP) to stimulate selected areas (designated as pre-activated areas as exemplified in *Figure 1—figure supplement 4* and *Video 3*) but only used 1–5% input of the 488-nm laser for 5–10 s. Next, we recorded the GCaMP6s-CAAX signals in the whole field.

For measurements of $Ca^{2+}$ influx using the red-emitting $Ca^{2+}$ sensor (R-GECO1.2 or R-CaMP2), a total of 300 ng DNA (100 ng mCh-LOVSoc and 200 ng $Ca^{2+}$ sensor) was transfected into HeLa or HEK293 T cells. The 561-nm laser source as used to excite red emission with blue light stimulation imposed as described above. Because the 561-nm laser cannot activate LOVSoc, both the ON and OFF phases of $Ca^{2+}$ fluctuation can be monitored by applying multiple dark-light cycles with an external pulsed LED light (470 nm at power intensity of 40 $\mu W/mm^2$) or using the 488-nm laser source from the Nikon A1R+ confocal microscope.

To monitor light-inducible NFAT nuclear translocation, we used a HeLa cell line stably expressing NFAT1$_{1-460}$-GFP. mCh-LOVSoc was transfected into this stable HeLa cell line and cells were imaged

24 hr posttransfection. A fixed blue light pulse of 30 s (40 μW/mm²) was applied to the transfected cells with the interpulse interval varying from 0.5, 1, 4, to 8 min. A total of 24 min time-lapse images were recorded and the GFP signal ratio (nuclear *vs* total GFP) was used to report the efficiency of NFAT activation. At least 15 cells were analyzed for each condition in three independent experiments.

Intracellular $Ca^{2+}$ measurements with Fura-2 were performed using our previous protocols (*Wang, 2014*; *Zhou et al., 2010a*; *2010b*; *2013*). Briefly, one day before imaging, HEK293 cells transiently expressing mCh-LOVSoc were seeded and cultured on cover slips. To load Fura-2 AM, cells were kept in the imaging solution with 0 mM $CaCl_2$ and 2 μM Fura-2 AM for one hour. The imaging solution consists of (mM) 107 NaCl, 7.2 KCl, 1.2 $MgCl_2$, 11.5 glucose, 20 HEPES-NaOH (pH 7.2), and 0 or 1 mM $CaCl_2$. Fura-2 signals were then obtained using a ZEISS oberserver-A1 microscope equipped with a Lambda DG4 light source (Sutter Instruments), Brightline FURA2-C-000 filter set (Semrock Inc.). Fura-2 fluorescence at 509 nm generated by 340 nm excitation light ($F_{340}$) and 380 nm light ($F_{380}$) was collected every two seconds, and intracellular $Ca^{2+}$ levels are indicated by $F_{340}/F_{380}$ ratio. To excite LOVSoc during light-on period, cells were continuously exposed to a 482 ± 9 nm light throughout each two-second interval immediately following the collection of every single $F_{380}$ and $F_{340}$. After 1 min photostimulation (470 nm, 40 μW/mm²), $Ca^{2+}$ concentrations in cells were determined by using a Fura 2 calcium imaging calibration kit (ThermoFisher Scientific) as we routinely did in earlier studies (*Wang, 2014*; *Zhou et al., 2010a*; *2010b*; *2013*). The resulting data collected with MetaFluor software (Molecular Devices) were then exported as txt file, analyzed with Matlab, and plotted using the Prism 5 software.

## NFAT-dependent luciferase reporter assay

HeLa cells were seeded in 24-well plates and transfected after reaching 40–50% confluence. mCh-LOVSoc, the firefly luciferase reporter gene (NFAT-Luc) and *Renilla* luciferase gene (pRL-TK) were co-transfected using Lipofectamine 3000. 24 hr posttransfection, cells were treated with PMA (1 μM) and/ or blue light (pulse of 30 s for every 1 min, 40 μW/mm²). Three duplicates were used for each treatment. After 8 hr, cells were harvested and luciferase activity was assayed by using the Dual Luciferase Reporter Assay System (Promega) on a Synergy luminescence microplate reader (BioTek, Winooski, VT, USA). *Renilla* luciferase is used as control reporter for counting transfected cells and normalizing the luminescence signals. The ratio of firefly to renilla luciferase activity was calculated and normalized to un-treated control group.

## Electrophysiological measurements

HEK EPC10 USB double patch amplifier controlled by Patchmaster software (HEKA Elektronik) was used for data collection. Conventional whole cell recordings were used to measure current in HEK293-ORAI1 stable cells transiently expressing mCh-LOVSoc as previously described (*Ma, 2015*). After the establishment of the whole-cell configuration, a holding potential of 0 mV were applied. A 50 ms step to −100 mV followed by a 50 ms ramp from −100 to +100 mV was delivered every 2 s. The intracellular solution contained (mM): 135 Cs aspartate, 6 $MgCl_2$, 10 EGTA, 3.3 $CaCl_2$, 2 Mg-ATP, and 10 HEPES (pH 7.2 by CsOH). The free $Ca^{2+}$ concentration in this pipette solution is estimated to be 100 nM based on calculations from http://www.stanford.edu/~cpatton/webmaxcS.htm. The extracellular solutions contained (mM): 130 NaCl (or N-methyl-D-glucamine, $NMDG^+$), 4.5 KCl, 20 $CaCl_2$, 10 TEA-Cl, 10 D-glucose, and 5 Na-HEPES (pH 7.4). A 10 mV junction potential compensation was applied to correct the liquid junction potential between the pipette solution relative to extracellular solution. Currents from at least 6 cells for each condition were collected. HEKA Fitmaster and Matlab 2014a software were used for data analysis.

## Isolation, culture, and retroviral transduction of mouse primary T cells

Platinum-E (Plat-E) retroviral packaging cell Line (Cell Biolabs, Inc, San Diego, CA) was maintained in Dulbecco modified Eagle medium (DMEM) supplemented with 10% fetal calf serum (FCS), 1% penicillin/streptomycin, and 1% glutamine. Plat-E cells were transiently transfected using Lipofectamine 3000 (Life Technologies) and retroviral stocks were collected twice at 24-hr intervals beginning 48 hr after transfection. Retrovirus-containing medium was centrifuged at 20 000 rpm for 2 hr at 4°C in a Beckman SW28 swinging bucket rotor lined with an Open-Top polyclear centrifuge tube (Seton,

Petaluma, CA). The retroviral pellet was resuspended in DMEM and retrovirus was titered by transduction of mouse T cells with serial dilutions of retrovirus in the presence of 8 µg/ml polybrene (EMD Millipore, Merck KGaA, Darmstadt, Germany). 48 hr posttransduction, percentage of infected cells was determined by flow cytometric analysis of EGFP expression. The titer was calculated by multiplication of the total number of EGFP-positive cells with the dilution factor of the retroviral supernatant.

Naive CD4[+] T cells were purified (>95% purity) by negative selection (Invitrogen) with Mouse Depletion Dynabeads (Life Technologies, Grand Island, NY) from RBC-lysed single-cell suspensions of pooled spleen and lymph nodes isolated from 6-week-old female C57BL/6 mice. For stimulation, purified CD4[+] T cells were cultured in DMEM supplemented with 10% heat-inactivated fetal bovine serum, 2 mM L-glutamine, penicillin-streptomycin, non-essential amino acids, sodium pyruvate, vitamins, 10 mM HEPES, and 50 µM 2-mercaptoethanol. Cells were plated at ~$10^6$ cells per ml in 6-well plates coated with anti-CD3 (clone 2C11, BioLegend, San Diego, CA, USA) and anti-CD28 (clone 37.51, BioLegend) (1 µg/ml each) by pre-coating with 100 µg/ml goat anti-hamster IgG (MP Biomedicals, Santa Ana, CA, USA). After 48 hr, cells were removed from the TCR signal and re-cultured at a concentration of $5 \times 10^5$ cells/ in T cell media supplemented with 20 U/ml recombinant human IL-2 (rhIL-2). For retroviral transduction, CD4[+] T cells were re-suspended in concentrated viral supernatant containing 8 µg/ml polybrene and rhIL-2 and centrifuged at 2,000 x g for 90 min at 32°C then put back to the incubator. On day 5–6, GFP+ cells were either left untreated (resting), or re-stimulated with PMA (15 nM) and ionomycin (0.5 µM), or subjected to blue light pulse for 6–8 hr (30 s pulse for every 1 min, 10–40 µW/mm$^2$), or treated with both PMA and blue-light pulse for 6–8 hr. Expression of cytokine production was assessed by real-time PCR and ELISA as described below.

## Real-time PCR analyses

Total RNA was isolated from transduced CD4[+] T cells and first-strand cDNA synthesis was performed using total RNA, oligo-dT primers and reverse transcriptase II according to manufacturer's instructions (Invitrogen). Real-time PCR was performed using the SYBR Green ER qPCR Super Mix Universal (Invitrogen) kit with specific primers using the ABI Prism 7000 analyzer (Applied Biosystems). The sequences of the primers are as follows,

Primers for mouse *Gapdh*
Forward: 5'-TTGTCTCCTGCGACTTCAACAG-3'
  Reverse: 5'-GGTCTGGGATGGAAATTGTGAG-3'

Primers for mouse interleukin 2 (*Il-2*)
Forward: 5'-TGAGCAGGATGGAGAATTACAGG-3'
  Reverse: 5'-GTCCAAGTTCATCTTCTAGGCAC-3'

Primers for mouse interferon gamma (*Ifn-γ*)
Forward: 5'-ATGAACGCTACACACTGCATC-3'
  Reverse: 5'-CCATCCTTTTGCCAGTTCCTC-3'

## Quantification of cytokines and insulin production by enzyme-linked immunoassay (ELISA)

Supernatants of transduced CD4[+] T cells were collected at indicated time after stimulation. Cytokine concentrations were measured by using the mouse IL-2 (OptEIA #555148, BD Biosciences Inc., San Jose, CA, USA) and IFN-γ ELISA kits (#88–7314, eBiosciences Inc., San Diego, CA, USA). ELISA assays were performed according to the manufacturer's instructions. In brief, 96-well plate was pre-coated with the capture antibody (1:500 in coating buffer) at 4°C overnight. On the next day, the plate was washed with PBS/0.1%Tween 20 and blocked with 1%BSA/PBS or ELISA/ELISPOT diluent buffer for 2 hr at room temperature (RT). Diluted supernatants and cytokine standards were then applied to the plate and incubated for 2 hr at RT. The plate was then washed and incubated with biotin-conjugated detection antibody (1:1000 in 1%BSA/PBS or ELISA/ELISPOT diluent buffer) for 1 hr at RT. Next, the plate was washed and incubated with poly-HRP streptavidin (1:5000 in diluent buffer, Thermo Scientific) for 30 min. The plate was finally washed and incubated with the

tetramethylbenzidine substrate solution (Sigma-Aldrich) and the reaction was stopped with 2 M $H_2SO_4$. For insulin reporter assay, $3 \times 10^5$ transfected HEK293T cells were cultivated in poly-L-lysine coated 24-well pates and starved in serum-free culture medium for 24 hr to ensure minimal activation of $Ca^{2+}$ dependent pathways. On the day of experiment, cells were washed with PBS and maintained in serum-/glucose-starved Krebs buffer (118 mM NaCl, 4.7 mM KCl, 1.2 mM $KH_2PO4$, 1.2 mM $MgSO_4$, 4.2 mM $NaHCO_3$, 2 mM $CaCl_2$, 10 mM HEPES and 0.1 mg/ml BSA, pH7.4) with or without light stimulation. Supernatants were collected for insulin ELISA detection using a human insulin ELISA kit (KAQ1251, Life Technologies) according to the manufacture's instructions. Absorbance of each well was recorded at 450 nm. The absorbance of the standard sample was used to construct the standard curve.

## Detection of activated caspase-1 and mature IL-1β

THP-1 cells from ATCC were maintained in RPMI-1640 medium containing 10% FBS and 0.05 mM 2-mercaptoethanol. Differentiated THP-1 cells were transduced with lentiviruses encoding LeGO-mCh-LOVSoc. THP-1 cells ($5 \times 10^5$) were seeded in 24-well plates and cultured overnight, followed by priming with 100 ng/mlLPS for 3 hr and stimulating with Nigericin (10 µM) for 6 hr with or without blue light stimulation. Medium from each well was mixed with 500 µl methanol and 125 µl chloroform, vortexed, and centrifuged for 5 min at 16,000 × g. The supernatant of each sample was removed and 500 µl methanol was added. Samples were centrifuged again for 5 min at 16,000 × g. Next, supernatants were removed and pellets were dried for 5 min at 50°C. 80 µl loading buffer was added to each sample, followed by boiling for 10 min prior to SDS-PAGE and immunoblot analysis with antibodies for the detection of activated caspase-1 (D7F10; Cell Signaling). The amounts of processed IL-1β were measured using a human IL-1β ELISA kits (R&D Systems) according to the manufacturer's instructions. Adherent cells in each well were lysed with the RIPA lysis buffer (50 mM Tris-HCl, pH 8.0, with 150 mM sodium chloride, 1.0% Igepal CA-630 (NP-40), 0.5% sodium deoxycholate, and 0.1% sodium dodecyl sulfate) with a protease inhibitors cocktail tablet (Roche), followed by immunoblot analysis to determine the cellular content of various proteins.

## Recombinant protein expression and purification

BL21 (DE3) *E.coli* cells (EMD Millipore) were transformed with plasmids encoding MBP-LOVSoc or GB1-ORAI1-CT, and grown at 37°C in LB medium with 100 mg/L of ampicillin. Protein expression was induced by the addition of 500 µM IPTG when $OD_{600}$ of the culture reached between 0.6 and 0.8. After IPTG induction, MBP-LOVSoc was incubated at 16°C for additional 6–8 hr, whilst GB1-ORAI1CT incubated at 37°C for 3–4 hr. Harvested cells were resuspended in 1X Phosphate Buffered Saline (PBS) and sonicated. The cellular debris was clarified by centrifugation. For $His_6$-tagged GB1-ORAI1-CT, the cell lysates were applied to $Ni^{2+}$-nitrilotriacetic acid (Ni-NTA)-agarose resin (Qiagen). Bound recombinant proteins were eluted in PBS containing 250 mM imidazole and 1 mM TCEP. MBP and MBP-LOVSoc were purified through affinity purification with amylose resin (New England Biolabs) and finally eluted by PBS buffer containing 25 mM maltose and 1 mM TCEP. The proteins were further purified by gel filtration on Superose 6 10/300 GL or Superdex 200 10/300 GL columns (GE Healthcare).

## Pulldown and coimmunoprecipitation (CoIP) experiments

For MBP pulldown assay, 400 µl 1 mg/ml of MBP (used as negative control) or MBP-LOVSoc was immobilized on 400 µl slurry of the amylose resin (New England Biolabs), and incubated with each 800 µg recombinant GB1-ORAI1-CT proteins in 1 ml PBS buffer containing 1 mM TCEP. The mixtures were divided into two groups: one group is constantly exposed to an external blue LED (470 nm, 40 µW/mm$^2$) for 4 hr at 4°C, and then followed by ten-time washing with PBS to minimize non-specific binding; whereas the other group was similarly treated except that all steps were performed in the dark. After extensive wash, the resin was finally mixed with 100 µl PBS and 4x SDS gel loading buffer, heated at 100°C for 10 min, and briefly centrifuged prior to gel electrophoresis. Samples were separated on 15% SDS-PAGE or 4–12% gradient NuPAGE. Bound proteins were visualized on SDS-PAGE after Coomassie Brilliant Blue R-250 staining.

For immunoprecipitation, HEK293 cells co-transfected with FLAG-ORAI1 and mCh-LOVSoc were lysed with 1x RIPA buffer containing protease inhibitor cocktails. Extracts were incubated for 1 hr

with anti-FLAG M2 affinity resin (A2220, Sigma) and the mixture was thoroughly washed with 1x RIPA buffer, denatured and eluted with 1x SDS sample buffer (62.5 mM Tris-HCl, pH 6.8, 2% SDS, 10% (v/v) glycerol, and 0.002% bromphenol blue). For light-stimulated groups, all steps were performed by exposing to an external blue LED (470 nm, 40 $\mu W/mm^2$).

## The UCNPs synthesis and modifications

All starting materials were obtained from commercial supplies and used as received. Rare earth oxides $Y_2O_3$ (99.9%), $Yb_2O_3$ (99.9%), $Tm_2O_3$ (99.9%), trifluoroacetic acid (99%), 1-octadecene (ODE) (>90%), oleic acid (90%), 1-ethyl-3-(3-dimethylaminopropyl) carbodiimide hydrochloride (EDC·HCl), N-hydroxysulfosuccinimide sodium salt (sulfo-NHS) and poly(acrylic acid) (PAA, $M_w$ 1,800) were purchased from Sigma-Aldrich. All other chemical reagents with analytical grade were used directly without further purification.

The size and morphology of UCNPs were determined at 200 kV at a JEM-2010 low to high- resolution transmission electron mircroscope (JEOL Inc., Peabody, MA, USA). The UCNP samples were dispersed in hexane and dropped on the surface of a copper grid for TEM test. The upconversion luminescence emission spectra were recorded on a Fluoromax-3 spectrofluorometer (Horiba Scientific, Irvine, CA, USA) that was equipped with a power adjustable collimated CW 980 nm laser. All the photoluminescence studies were carried out at room temperature.

The $\beta$-$NaYF_4$:Yb,Tm core UCNPs were prepared using a modified two-step thermolysis method (*Mai et al., 2006*). In the first step, the $CF_3COONa$ (2 mmol) and required $Ln(CF_3COO)_3$ (0.5 mmol in total) precursors were mixed with oleic acid (5 mmol), oleyl amine (5 mmol) and 1-octadecene (10 mmol) in a two-neck reaction flask. The mol-percentage of $Tm(CF_3COO)_3$ was fixed at 0.5%, Yb $(CF_3COO)_3$ was employed in 80%, and $Y(CF_3COO)_3$ was used of 19.5%. The slurry mixture was heated to 110°C in order to form a transparent solution. This was followed by 10 min of degassing to remove the oxygen and water. The flask was then heated to 300°C at a rate of 15°C per min under dry argon flow, and remained at 300°C for 30 min. The $\alpha$-$NaLnF_4$ intermediate UCNPs were acquired by cooling down the reaction solution to room temperature, followed by centrifugation with excessive ethanol. In the second step, the $\alpha$-$NaYF_4$:Yb, Tm UCNPs were re-dispersed in oleic acid (10 mmol) and 1-octadecene (10 mmol) along with $CF_3COONa$ in a two-neck flask. After degassing at 110°C for 10 min, the flask was heated to 325°C at a rate of 15°C per min under dry argon flow, and remained at 325°C for 30 min. The $\beta$-$NaYF_4$:Yb,Tm UCNPs were then centrifugally separated from the cooled reaction media and suspended in 10 ml of hexane as the stock solution for further use.

In the thermolysis reaction, as-synthesized $\beta$-$NaYF_4$:Yb, Tm UCNPs served as crystallization seeds for the epitaxial growth of undoped $\beta$-$NaYF_4$ shell. Typically, a stock solution of $\beta$-$NaYF_4$:Yb, Tm UCNPs (5 ml, *ca.* 0.26 µmol/L core UCNPs) was transferred into a two-neck flask and hexane was sequentially removed by heating. Then $CF_3COONa$ and $Y(CF_3COO)_3$ (0.5 mmol) were introduced as UCNP shell precursors with oleic acid (10 mmol) and 1-octadecene (10 mmol). After 10 min of degassing at 110°C, the flask was heated to 325°C at a rate of 15°C/min under dry argon flow and was kept at 325°C for 30 min. The products were precipitated by adding ethanol to the cooled reaction flask. After centrifugal washing with hexane/ethanol, the core/shell UCNPs were re-dispersed in 10 ml of hexane for further use.

## The synthesis of UCNPs-Stv

The hydrophobic UCNPs were first treated by surface ligand exchange using a modified literature method (*Dong et al., 2011*). Briefly, nitrosonium tetrafluoroborate ($NOBF_4$, 0.20 g) was dissolved in dimethylformamide (DMF, 5 ml), and $\beta$-core/shell UCNPs in hexane stock solution (1 ml) was added, followed by 4 ml hexane and 2 hr of stirring at room temperature. Then $BF_4^-$ capped UCNPs were precipitated by adding isopropanol (5 ml), and purified by 2 cycles of centrifugal wash with DMF. Subsequently, all UCNPs precipitates were dispersed in poly(acrylic acid)/DMF (PAA, $M_w$ 1800, 10 mg/ml, 5 ml) solution to replace surface $BF_4^-$ by PAA. After overnight incubation, the PAA coated $\beta$-$NaYF_4$:Yb,Tm/$NaYF_4$ UCNPs were purified by centrifugal wash with deionized (DI) water.

The streptavidin and zwitterion ligands (*Muro et al., 2010*) were conjugated to UCNPs-PAA surface by EDC (1-Ethyl-3-(3-dimethylaminopropyl)-carbodiimide) coupling approach. Generally, 50 mg hydrophilic PAA-coated UCNPs in 5 ml DI water were activated by EDC (50 mg) and NHS (10 mg) to form succinimidyl ester. After stirring at room temperature for 2 hr, the nanoparticles were collected

by centrifugation followed by washing with DI water. The generated nanoparticles were then re-dispersed into 5 ml DI water, followed by adding 150 µg streptavidin and the mixture was further stirred at room temperature for 4 hr. Next, 100 mg zwitterion ligand was introduced to the solution. After overnight stirring at room temperature, the UCNPs-Stv were purified by washing with DI water, centrifugation and dispersion in DI-water for further use.

### In vitro LOV2 domain activation mediated by UCNPs-Stv following NIR light stimulation

5 ml LOV2 or MBP-LOVSoc proteins were concentrated to 0.5 ml at a concentration of 50–100 µM using centrifugal filter devices with a cutoff of 10 kDa. The UV-Vis spectra were recorded with a Shimadzu or Nanodrop 2000 spectrophotometer (Thermo Scientific, Waltham, MA, USA). The absorbance was recorded before and after the introduction of UCNPs-Stv. 10 mg of UCNPs-Stv was added to make a final concentration of 20 µg/µl. The mixed solution was then transferred to a thin glass tube (with a diameter comparable to the CW laser spot) and subjected to 980 nm CW laser excitation (15 mW/mm$^2$) for 1 min. The control sample was exposed to blue light (470 nm, 40 µW/mm$^2$) for 1 min. After light stimulation, the absorbance was monitored every 30–300 s till the LOV2 domain fully returned to its dark state.

### Fourier transform infrared spectroscopy (FT-IR)

20 mg of UCNPs with different surface modifications were mixed with 100 mg KBr, and then grounded into fine powder in a mortar. A piece of pre-cut cardboard was placed on top of a stainless steel disk and the cutout hole was filled with the finely ground mixture. A second stainless steel disk was put on top and the sandwich disks were transferred onto the pistil in the hydraulic press to obtain a homogenous and transparent film. The samples were then inserted into the IR sample holder for analysis. Black background (KBr film only) was subtracted from the corresponding spectrum.

### Quantification of upconversion quantum efficiency

The upconversion quantum efficiency (QE) is used to precisely measure the upconversion ability of the characterized materials, which is defined as the fraction of the absorbed photons that successfully employed to generate upconversion emission. The upconversion QE was calculated based on the following equation: QE=i*QY; where QY represents the quantum yield and i equals to 3 as Tm$^{3+}$ excited state produces three-photon luminescence at 480 nm (from $^1D_2$ state to $^3H_6$ state). The upconversion QY was first measured on a relative basis, using a known QY (3.2%) sample of α-NaYF$_4$:Yb,Er @CaF$_2$ as a standard (*Punjabi et al., 2014*). The following equation was used to calculate the QY:

$$QY_{Sample} = QY_{ref} \left( \frac{E_{sample}}{E_{ref}} \right) \left( \frac{A_{ref}}{A_{samples}} \right)$$

where (E) is the integrated emission intensity at 480nm, (A) is the absorption at 980 nm. The upconversion QE of the 40-nm $\beta$-NaYF$_4$:Yb,Tm@$\beta$-NaYF$_4$ UCNPs in the blue region was determined to be 2.7% at the power density of 10 W/cm$^2$.

### Fluorescence imaging with UCNPs

#### Imaging of UCNPs-Stv targeted to HeLa cells transfected with mCh-ORAI1$^{StrepTag}$

Transfection reagent (100 ng of pcDNA-mCh-ORAI1$^{StrepTag}$ in 50 µL opti-MEM) was mixed with lipofectamine solution (2 µL of lipofectamine in 50 µL opti-MEM). 5 min later, the plasmid mixture was added into petri dish with 0.1 million HeLa cells. The cells were incubated with transfection reagent in opti-MEM for 4 hr, returned to DMEM and allowed for further growth of 16 hr. 100 µL of UCNPs-Stv PBS solution was introduced into the cell culture media and incubated for 2 hr, followed by washing and re-addition of opti-MEM for imaging. For imaging, Images were recorded on a LSM7 MP microscope (Zeiss) equipped wavelength adjustable coherent lasers with 60× water immersion objective lens. mCherry was excited at 740 nm and emission was detected from 610 to 650 nm. While UCNPs was excited at 980 nm and emission was detected from 450 to 500 nm.

## Imaging of transfected HeLa cells in the presence of UCNPs-Stv

HeLa cells were cotransfected with a total of 500 ng DNA (200 ng of pTriEX-mCh-LOVSoc, 200 ng of pcDNA3.1-mCh-ORAI1-StrepTag, 100 ng of pGP-CMV-GCaMP6S-CAAX or 100 ng NFAT1$_{1-460}$-GFP in opti-MEM) as described above. 16 hr posttransfection and 2 hr prior to imaging, 20 mg of UCNPs-Stv PBS solution was introduced into the cell culture media. For imaging, Petri dish was mounted on a Leica TCS SP 2 confocal microscope equipped with a 63×oil objective. mCherry was excited at 590 nm and emission was detected from 610 to 670 nm. A 488-nm laser with minimum power was used to acquire GFP signals whilst a 590-nm laser was applied to acquire mCherry signals. All images were collected at a scanning rate of 400 Hz. 980 CW laser was introduced into the system with a power density of 15 mW/mm$^2$, and each irradiation takes 5–10 s. The relatively slow onset of Ca$^{2+}$ influx and NFAT nuclear translocation provided us a time window to quickly capture the green signals without noticeably activating LOVSoc during image acquisition. This allows us to confidently apply NIR light to monitor Ca$^{2+}$ influx and NFAT nuclear translocation.

## Bioluminescence and thermal imaging

HeLa cells were transfected with NFAT-Luc with and without the Opto-CRAC construct LOVSoc, as indicated. 48 hr after transfection, $5 \times 10^5$ cells suspended in 200 μL DMEM with 1 μM PMA were mixed with 10 mg UCNPs-Stv, then implanted i.v. into BALB/c mice (female; 4–8 weeks; injected position: upper thigh, as indicated in red circle; from Jackson Laboratory). The hairs on the back of the mice were shaved, whilst the hairs on the belly remained unshaved. The implanted regions were subject to 980 nm CW laser irradiation (50 mW/mm$^2$, 30 sec every 1 min for a total of 25 min), during anesthesia using ketamine/xylazine (100 mg/kg, 10 mg/kg, i.v.). Five hours later, the cells implanted area was injected with D-luciferin (s.c., 100 μL, 15 mg/ml in PBS) and imaged 20 min later with an IVIS-100 in vivo imaging system (2-min exposure; binning = 8). Luciferase luminescence was plotted as false color with rainbow-scale bar set as the same for all acquired images. For thermal imaging, BALB/c mice were immobilized and exposed to 50 mW/mm$^2$ 980 nm CW laser under the same condition as we carried out for the in vivo luciferase experiment. Images at two-minute intervals were taken by a thermal imaging camera (FLIR Instruments).

## Ex vivo cross-presentation assay and OT-I T-cell activation

To obtain murine bone marrow-derived dendritic cells (DCs), bone marrow cells were washed out of the femurs of adult mice in RPMI-1640 using a syringe and a 25-gauge needle and depleted of red blood cells. Bone marrow cells ($5 \times 10^5$ cells/well) in 6-well plate were cultured in RPMI-1640 containing 2 mM-L-glutamine,100 IU/ml penicillin,100 mg/ml streptomycin, 10% FCS,50 μM $\beta$-ME,20 ng/ml GM-CSF and 200 IU/ml IL-4 for dendritic cell differentiation. Bone marrow cells were transduced with MSCV expressing viral vector pMIG-mCh-LOVSoc-IRES-ORAI1$^{StrepTag}$ on day 3 at MOI of 20 for 6 hr. Next, 75% of the media and non-adherent cells were removed and replaced with fresh culture medium. On day 5, transduced DCs were gently dislodged and pulsed for 3 hr at 37°C with 2 μg/ml OVA$_{257-264}$ peptide (GenScript) and 1 mg/ml UCNP-Stv nanoparticles. Cells were then washed to remove unattached peptide and nanoparticles. To generate OT-I CD8 T cells, spleens and lymph nodes (LN) of OT-1 Rag1$^{-/-}$ mice (purchased from the Jackson Laboratory) were pressed through a 70 μm cell strainer (BD Falcon). Untouched CD8 T cells were sorted by using mouse CD8 T Cell Isolation Kit (Miltenyi Biotec). $2 \times 10^4$ irradiated peptide loaded UCNPs-Stv/OVAp Opto-CRAC DCs were seeded in triplicates in 96-well U-bottom plates containing $5 \times 10^4$ purified OT-I CD8 T cells in a total volume of 200 μl and co-cultured for 5 days with or without NIR light stimulation for 16h (1 min pulse, 15 mW/mm$^2$). T cell proliferation was determined by labeling cultured cells with [$^3$H] thymidine at a concentration of 1 μCi/μL for 16 hr and the radioactivity was measured using a liquid scintillation counter (PerkinElmer). To detect DCs maturation and migration, NIR-stimulated or unstimulated UCNPs-Stv Opto-CRAC DCs were stained with FITC-CD11c, PE-MHC-I, APC-MHC-II, PE-CD86 and PE-CCR7 and then subjected to flow cytometry analysis 3 days post-transduction. For intracellular IFN-γ staining, OT-I CD8 T cells were incubated with UCNPs-Stv/OVAp Opto-CRAC DCs for 6 hr at 37°C in the presence of GolgiStop (monensin) (BD Pharmingen). Cells were then stained with surface marker using APC-CD8a antibody for 15 min on ice and permeabilized using cytofix/cytoperm (BD Biosciences) for 30 min on ice. Permeabilized cells were resuspended in BD

Perm/Wash buffer (BD Biosciences) and stained with PE-anti-IFN-γ antibody for 20 min. Samples were run on a BD LSRII Flow Cytometer and analyzed by BD FACSDiva software.

## Adoptive cell transfer in murine B16-OVA melanoma models

B16-OVA is an OVA-transfected clone derived from the murine melanoma cell line B16 (ref. 35). B16-OVA cells were cultured and maintained in Dulbecco's modified Eagle medium (HyClone) supplemented with 10% heat-inactivated fetal bovine serum (FBS), 100 IU/ml penicillin, 100 mg/ml streptomycin under 37°C in 5% $CO_2$. B16-OVA cells ($1 \times 10^6$) were injected *s.c.* into the flank region or *i.v.* via tail vein of *Rag1*$^{-/-}$ mice (purchased from the Jackson Laboratory) (*Overwijk and Restifo, 2001*). 3 days later, mice were injected *i.v.* with $2 \times 10^5$ Opto-CRAC DCs treated with UCNPs-Stv and the surrogate tumor antigen OVA$_{257-264}$. $1.5 \times 10^6$ OT-I T cells labeled with CellTrace far red CFSE were *i.v.* injected into tumor-bearing mice. Briefly, cells were incubated at $1 \times 10^6$ cells/ml in CFSE at a final concentration of 1 µM for 20 min at room temperature. The labeling reaction was stopped by adding the same volume of FBS. Recipient *Rag1*$^{-/-}$ mice were subjected to the excitation of NIR laser (8 hr per day, 0.5–1 min ON/OFF pulse, 30 mW/mm$^2$) or shielded from NIR (control group) for 6 days to stimulate Opto-CRAC DC maturation, with the initial two days concentrating more on areas nearby the draining lymph nodes of restricted mice. For in vivo T cell proliferation, spleen and draining popliteal and inguinal LNs were harvested and injected with collagenase D (1 mg/ml; Boehringer-Mannheim, Mannheim, Germany) in RPMI and 10% FBS for 20 min at 37°C. Digested LN or spleen were filtered through a stainless-steel sieve, and the cell suspension was washed twice in PBS and 5% FBS. CFSE-labeled OT-I CD8 T cells were analyzed by flow cytometry as described above. Tumor growth was measured at indicated time points using calipers shown in growth curve using the equation of $V = L \times W^2/2$. Lungs were isolated and tumor foci of lung melanomas were counted from tumor-bearing mice shielded or exposed to NIR pulse from day 3–9 after B16-OVA tumor cell injection.

## Histology analyses

Hela cells and UCNPs were subcutaneously implanted into upper thigh of BALB/c mice, followed by 980 nm CW laser irradiation (50 mW/mm$^2$, 30 sec every 1 min for a total of 25 min), during anesthesia using ketamine/xylazine (100 mg/kg, 10 mg/kg, i.v.). Two weeks after subcutaneous implantation, mice were sacrificed and tissue samples under skin at the injection position were collected. Routine Hematoxylin and Eosin staining (H&E) was performed by University of Massachusetts Medical School morphology core.

## Data analyses

The fluorescence images were analyzed with the NIS-Elements imaging software (Nikon) or the Image J package (NIH) with the intensities plotted using the GraphPad Prism 5 graphing and statistical software. The mean lifetime of fluorescence signal change was calculated with a single exponential decay equation $F(t)=F(0)*e(-t/\tau)$. Quantitative data are expressed as the mean and standard deviation of the mean (s.e.m.) unless otherwise noted. Paired Student's *t*-test was used throughout to determine statistical significance. *P<0.05; **P<0.01; ***P<0.001, when compared to control or WT.

## Acknowledgements

We thank Anjana Rao for her advice on T cell-related experiments and critical feedback. We are grateful for Dr. Klaus Hahn for sharing the PA-Rac1 plasmid. This work was supported by the National Institutes of Health grants (R01GM112003 to YZ, R01MH103133 to GH, RO1AI084167 and RO1GM110397 to PH), the Cancer Prevention Research Institute of Texas (RR140053 to YH), the Special Fellow Award from the Leukemia and Lymphoma Society (LLS3013-12 to YZ), the Human Frontier Science Program (to GH), the China Scholarship Council (to JJ), the National Natural Science foundation of China (NSFC-31471279 to YW) and by an allocation from the Texas A&M University Health Science Center Startup Fund (YZ).

## Additional information

### Funding

| Funder | Grant reference number | Author |
|---|---|---|
| National Institutes of Health | R01GM112003 | Yubin Zhou |
| National Institutes of Health | R01MH103133 | Gang Han |
| National Institutes of Health | R01AI084167 | Patrick Hogan |
| National Institutes of Health | R01GM110397 | Patrick Hogan |
| The Cancer Prevention Research Institute of Texas | RR140053 | Yun Huang |

The funders had no role in study design, data collection and interpretation, or the decision to submit the work for publication.

### Author contributions

LH, GH, YZho, YZha, PT, YH, Conception and design, Acquisition of data, Analysis and interpretation of data, Drafting or revising the article; GM, YH, PGH, ZL, SZ, XW, JJ, SF, LZ, YW, Conception and design, Acquisition of data, Analysis and interpretation of data

### Ethics

Animal experimentation: Mice-related experiments were approved by Institutional Animal Care and Use Committees of Institute of Biosciences and Technology, Texas A&M University Health Science Center (#12044 and #2014-0228-IBT; Houston, TX, USA; Animal Welfare Assurance Number A3893-01) and University of Massachusetts Medical School (#A-2512-15, Worcester, MA, USA; Animal Welfare Assurance Number A3306-01).

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
