## [Decision Letter]

Thank you for submitting your work entitled "Near-infrared photoactivatable control of Ca^2+^ signaling" for peer review at *eLife*. Your submission has been evaluated by John Kuriyan (Senior editor) and three reviewers, one of whom, Richard Aldrich, is a member of our Board of Reviewing Editors, and another is Murali Prakriya.

The reviewers have discussed the reviews with one another and the Reviewing editor has drafted this decision to help you prepare a revised submission.

Summary:

He et al. integrated several techniques to a create photoactivatable system called "Opto-CRAC" that controls Ca^2+^ influx into cells through CRAC channels. The high Ca^2+^ selectivity of CRAC channels is a clear advantage over the current non-selective channelrodopsin-based optogenetic systems and the authors use this as a central justification for developing methods that would directly elicit Ca^2+^ influx instead of relying on membrane depolarization (in the channelrodopsin-based optogenetic systems) to activate endogenous Ca^2+^ channels. The study develops a protein construct with LOV2 (a photoswitch that responds to blue-light) fused to a STIM1 C-terminal fragment. Using a HEK293 cell line that stably expresses Orai1 transfected with their chimeric construct, they demonstrate tight control of Ca^2+^ influx using blue light. They also showed that NFAT-translocation, which is a downstream effect of CRAC channel activation, correlates with the frequency of the blue light-mediated Ca^2+^ pulses. In addition, the study pairs the opto construct with lanthanide-doped up conversion nanoparticles (UCNP) that absorb in the near-infrared wavelengths and emit blue light. This allowed control Ca^2+^ influx and NFAT-translocation in cells using near-infrared light, which is necessary in developing a system that can penetrate tissues in an in vivo system. Finally, as a proof-of-concept experiment, HeLa cells expressing the Opto-CRAC system are grafted subcutaneously into the flanks of mice and the manuscript demonstrates changes in NFAT-luciferase expression using near-infrared light in these cells under the skin. A light activated Ca^2+^ channel could be tremendously useful for applications ranging from studies of basic biology of CRAC channels, local activation of CRAC channels in microdomains within cells, and in vivo interrogation of immune and other cells.

Essential revisions:

While there is considerable enthusiasm for the successful development of Opto-CRAC as a tool, particularly for use in non-excitable cells, the reviewers feel that there remains much to be done to eliminate current problems that make it unfeasible. While we recognize that this is essentially a proof of concept paper for a new technique, we feel that the following issues must be adequately addressed if the paper is to be accepted for publication.

1) In the current form, the system's limitations are substantial, even at this proof-of-concept stage. Optogenetic stimulation offers two key advantages: the requirement for a single protein and rapid kinetics. On the other hand, channelrhodopsin use is somewhat constrained by the need for blue light, which necessitates optical fiber implantation and hinders concurrent functional imaging, by its rapid inactivation, and by its limited single channel conductance. While Opto-CRAC offers significant advantages in being more calcium selective than channelrhodopsin based methods, It functions on the order of seconds to minutes, far slower than channelrhodopsin, limiting its usefulness to slower signaling processes, like many of those occurring in non-excitable cells. Although only a single protein is required for visible light stimulation, the resulting conductances are low, as judged by dynamic ranges of the employed genetically-encoded calcium indicators. In fact, it seems that the conductances are smaller than those produced by channelrhodopsin despite much longer stimulus duration. This may be due to the reliance on native ORAI channels – unlike many existing stimulation methods, where the heterologous activator is in excess, endogenous channels limit maximal conductance and make that conductance highly cell type-dependent. Moreover, induction of gene expression is predicated on concurrent treatment of cells with phorbol ester, a potent carcinogen. For these reasons, Opto-CRAC is far from being useful in animal studies.

2) However, the authors promote Opto-CRAC as a non-invasive NIR deep tissue cellular stimulator of non-excitable cells. Their assertion that no tools exist for this purpose is inaccurate, since multiple visible light-independent methods, including DREADDs, have proven quite effective outside the brain. For NIR use in culture, the authors assemble a far more elaborate system than described for visible light, consisting of engineered STIM and ORAI proteins, streptavidin-coated UCNP beads and PMA (for gene induction). For in vivo use, they implant cells expressing the encoded components that have been pre-treated with beads subcutaneously, hardly a non-invasive procedure.

3) Critically, the properties of this four-part (and not genetic) NIR system are inadequately described: NIR-dependent calcium entry is shown (as a function of GCaMP fluorescence), but not measured; NFAT translocation to the nucleus is demonstrated, but no gene expression data is provided.

4) How does one reconcile the complexity and limited sensitivity of Opto-CRAC, including the requirement for UNCP beads, with its intended application to modulate calcium in cells of the immune and hematopoietic systems? Are stem cells propagated and pre-treated ex vivo to be injected into the bone marrow or thymus? Will NIR, which elevated reporter expression in subcutaneous cells, have any impact on those tissues? No feasibility testing is described.

5) The one area where UNCPs might have a real impact as NIR light transducers is for activating channelrhodopsin (in fact, the inability of ChR to be gated by long-wave light is given as an explicit motivation for UNCP development). Injected locally and recruited to genetically targeted cells that express channelrhodopsin, UNCP could enable NIR ChR gating. Surprisingly, this potentially exciting application is not explored.

6) Is the relatively slow time scale due to properties of ORAI channels, rather than a limitation of the light induction by STIM1-LOV? If so, the comparison to channelrhodopsin should probably be tempered. At least some discussion, and perhaps some experiments should be included as to which protein is limiting (both in terms of maximal conductance and kinetics).

7) STIM1 has other targets besides Orai1 channels including other Orai isoforms, voltage-activated Ca^2+^ channels (which it inhibits), and even TRP channels. How the optically responsive STIM1 (LOVSoc) fits into this larger framework of potential targets is unclear. Since other channels may be engaged by STIMs, the authors should examine calcium selectivity by testing for other ions in cells that have additional endogenous channels, as opposed to using fibroblasts stably expressing ORAIs.

8) The vector size appears small enough for viral gene delivery, but it is unclear how UCNPs can be delivered. In addition, if the UCNPs binds to off-targets, that could wreak havoc with the high-energy blue light in the body.

9) Regarding the UNCPs: the conversion here is from low energy to high, which could be highly inefficient. Energies of excitation and emission at different wavelengths should be included.

---

## [Author Response]

Essential revisions:

*While there is considerable enthusiasm for the successful development of Opto-CRAC as a tool, particularly for use in non-excitable cells, the reviewers feel that there remains much to be done to eliminate current problems that make it unfeasible. While we recognize that this is essentially a proof of concept paper for a new technique, we feel that the following issues must be adequately addressed if the paper is to be accepted for publication. 1) In the current form, the system's limitations are substantial, even at this proof-of-concept stage. Optogenetic stimulation offers two key advantages: the requirement for a single protein and rapid kinetics. On the other hand, channelrhodopsin use is somewhat constrained by the need for blue light, which necessitates optical fiber implantation and hinders concurrent functional imaging, by its rapid inactivation, and by its limited single channel conductance. While Opto-CRAC offers significant advantages in being more calcium selective than channelrhodopsin based methods, It functions on the order of seconds to minutes, far slower than channelrhodopsin, limiting its usefulness to slower signaling processes, like many of those occurring in non-excitable cells. Although only a single protein is required for visible light stimulation, the resulting conductances are low, as judged by dynamic ranges of the employed genetically-encoded calcium indicators. In fact, it seems that the conductances are smaller than those produced by channelrhodopsin despite much longer stimulus duration. This may be due to the reliance on native ORAI channels – unlike many existing stimulation methods, where the heterologous activator is in excess, endogenous channels limit maximal conductance and make that conductance highly cell type-dependent.*

We understand the reviewers’ perspectives with regard to difference between ChR2-based tools and our Opto-CRAC system. Nonetheless, we would like to stress that while ChR2 is most useful in neuroscience with respect to the control of neuronal excitability, our study aims to expand the repertoire of optogenetic tools, with the goal of photo-manipulating cellular events that occur at a relatively slower time scale (seconds to minutes). Here we primarily focus on cells of the immune system to present and advocate the concept of optogenetic immunomodulation. We have demonstrated the significance and wide adaptivity of our tool for different cellular systems. In a dozen of non-excitable cell types that we have tested thus far, the expression of LOVSoc alone was clearly sufficient to elicit Ca^2+^ influx at endogenous levels of ORAI1 and caused cytosolic Ca^2+^ elevation of up to 500-800 nM (Figure 1—figure supplement 3). The magnitude and relatively slow kinetics of photoactivated Ca^2+^ influx largely mimic the physiological events upon engagement of immunoreceptors (as reviewed Prakriya and Lewis, 2015). Although the unitary conductance of CRAC channel is known to be extremely low (~9 fS in 2 mM extracellular Ca^2+^ or 24 fS in isotonic Ca^2+^ solution in T cells; compared to 4-10 pS for voltage-gated Ca^2+^ channels; Prakriya and Lewis 2015), the sustained Ca^2+^ entry (up to minutes) through native ORAI1 channels is sufficient to trigger downstream effectors. The small unitary conductance of the CRAC channel, coupled with its high Ca^2+^ selectivity (with a Ca^2+^: Na^+^ permeation ratio of over 1000; among the most selective Ca^2+^ channels known), is speculated to limit membrane depolarization, thereby reducing the energy requirement to pump out Na^+^ during sustained Ca^2+^ influx and enhancing the specificity of downstream effector function (Prakriya and Lewis 2015). The physiological hallmarks of the Ca^2+^/NFAT pathway can be fully recapitulated by our Opto-CRAC system.

In the revised manuscript, we have demonstrated the application of Opto-CRAC to i) photo-induce Ca^2+^-dependent gene expression, including luciferase and insulin production driven by Ca^2+^-dependent transcriptional factors (Figure 2); ii) photo-trigger T cell activation (Figure 2); iii) photo-amplify inflammasome response mediated by macrophages (Figure 2); and iv)photo-stimulate antigen presentation by dendritic cells to boost cytotoxic T cell activities both ex vivo and in vivo (Figure 4).

We respectfully disagree with the comment that the system is far from being useful in animal studies. For in vivo application, we find that our system is most compatible with adoptive cell transfer experiments or therapies, which are widely used during immunotherapies in both basic research and the clinic settings. As exemplified in Figure 4, by boosting DC-mediated antigen presentation and subsequent activation of cytotoxic T cells, we have successfully used our NIR Opto-CRAC system to photo-tune anti-tumor response in a mouse model of melanoma to suppress tumor growth.

*Moreover, induction of gene expression is predicated on concurrent treatment of cells with phorbol ester, a potent carcinogen. For these reasons, Opto-CRAC is far from being useful in animal studies.*

T cell activation requires the CD28 co-stimulatory receptor to activate AP-1 in order to cooperate with NFAT and drive a productive immune response. Please note that we only used phorbol ester to mimic the co-stimulation pathway, in vitro, for the purpose of a simpler and convenient experimental setup, but it is not a must. For example, phorbol ester can be readily replaced by anti-CD28 antibody to trigger co-stimulatory signals. In addition, to drive light-inducible Ca^2+^-dependent gene expression, we can circumvent the use of phorbol ester by fusing target genes (e.g., insulin) downstream of multiple Ca^2+^-response elements (Stanley et al., 2012), as shown in Figure 2. Moreover, in the revised manuscript, rather than repeating similar experiments in T cells, we focused on other types of immune cells that obviate the need for co-stimulatory pathways. As can be seen in Figure 2 and Figure 4, light-induced Ca^2+^ elevation in macrophages and dendritic cells, without the use of phorbol ester, is sufficient to induce the downstream events to amplify inflammatory response or to promote DC antigen presentation.

*2) However, the authors promote Opto-CRAC as a non-invasive NIR deep tissue cellular stimulator of non-excitable cells. Their assertion that no tools exist for this purpose is inaccurate, since multiple visible light-independent methods, including DREADDs, have proven quite effective outside the brain.*

After further studies about this point, we have followed the reviewers’ suggestion and toned down our claim about the non-existence of such tools.

*For NIR use in culture, the authors assemble a far more elaborate system than described for visible light, consisting of engineered STIM and ORAI proteins, streptavidin-coated UCNP beads and PMA (for gene induction). For in vivo use, they implant cells expressing the encoded components that have been pre-treated with beads subcutaneously, hardly a non-invasive procedure.*

The complexity of the NIR system has been substantially reduced by adopting the following strategies in the revised manuscript:i) a bicistronic IRES (internal ribosomal entry site) vector co-expressing engineered ORAI1 and LOVSoc, which is routinely used in the transduction of cells of the immune system and in adoptive immunotherapies; or ii) the protein co-expression system based on a self-cleaved 2A peptide (Figure 1—figure supplement 5 and Figure 4). Thus, the system now requires only a single plasmid construct, in addition to functionalized UCNPs.

For in vivo application, we find that our system is most compatible with adoptive cell transfer experiments or adoptive immunotherapies, which are widely used in both basic research and the clinic settings. We acknowledge that, like any adoptive cell therapy, our system also requires ex vivo expansion of engineered immune cells, preincubation with UCNPs, as well as re-injection into the bodies (as illustrated in Figure 4). In order to mitigate the reviewers’ concern, we have removed the word “non-invasive” throughout the text.

*3) Critically, the properties of this four-part (and not genetic) NIR system are inadequately described: NIR-dependent calcium entry is shown (as a function of GCaMP fluorescence), but not measured; NFAT translocation to the nucleus is demonstrated, but no gene expression data is provided.*

The NIR-dependent calcium response curve was not shown in the original paper for GCaMP6 due to the spectroscopic conflicts between NIR-excited emission (centered around 470 nm) and GFP excitation laser source (488 nm), the latter of which is required to record GCaMP6 signals. We have to use very low light input (<1 μW/cm^[3]^, Figure 1—figure supplement 3) to avoid direct activation of LOVSoc by the GFP-channel laser, so that we can ascribe the contribution solely to NIR-to-blue light stimulation. In the revised manuscript, we presented the NIR-induced Ca^2+^ response curve by using R-GECO1.2 as readout to circumvent this complication (Figure 3). In addition, as suggested by the reviewers, we have included NIR-inducible gene expression data by using IFN-γ production in sorted primary CD4^+^ T cells as a physiologically-relevant example (Figure 3).

*4) How does one reconcile the complexity and limited sensitivity of Opto-CRAC, including the requirement for UNCP beads, with its intended application to modulate calcium in cells of the immune and hematopoietic systems? Are stem cells propagated and pre-treated ex vivo to be injected into the bone marrow or thymus? Will NIR, which elevated reporter expression in subcutaneous cells, have any impact on those tissues? No feasibility testing is described.*

Please refer to our response to Comments 1 and 2 with regard to our efforts to reduce this complexity. As pointed out in our response to Comment 2, the images presented in Figure 3(original Figure 2) have a relatively low resolution/intensity because they have to be acquired under quite low laser input (< 1 μW/cm^[3]^) in order to avoid direct activation of LOVSoc by GFP excitation (~488 nm), the latter of which is required to record signals from GCaMP6 or GFP-NFAT. We now use the red-emitting R-GECO1.2 fluorescence as readout so as to avoid such complication. The consequent Ca^2+^ signals generated by our blue and NIR light are at comparable levels (Figure 3Evs. Figure 1—figure supplement 3). Please note that the Y-axis of GCaMP6 or R-GECO1.2 response has been shown as the ratio (not percentage) of ΔF over F_0_ (the value of 1 means 100% change in the GCaMP6s signals; or the Ca^2+^ signal change was doubled).

Most importantly, we have demonstrated NIR-induced Ca^2+^-dependent physiological responses both in vitro and in vivo (Figure 3–Figure 4), which clearly attests to the high feasibility and compatibility of our system with animal studies. We have carried out two proof-of-concept experiments to demonstrate the in vivo application of our system (Figure 3 and Figure 4). The elevated expression of the reporter gene luciferase (Figure 3) in subcutaneous cells did not induce tissue damages (as indicated by histological sections of the positions after 14 days of ectopic injection in mice; Figure 3—figure supplement 3), nor did NIR light generate a significant local heating effect (Figure 3—figure supplement 3).

As pointed out by the reviewers, our system is most compatible with adoptive cell therapies (such as adoptive T cell or DC vaccine-based immunotherapy), which have been gaining wide popularity in the treatment of cancer. Although the proposed experiment (bone marrow transplant) by the reviewers is quite attractive, it would take at least 4-6 months to establish the system and get the related animal protocols approved by IACUC committees. Given the limited 2-month window for revision, we performed a similar proof-of-concept experiment using engineered dendritic cells to boost CTL-mediated anti-tumor response in mouse melanoma models (via subcutaneous implantation or *i.v.* injection), which can be conveniently generated within 2-3 weeks (Figure 4). Results from our in vivo studies clearly indicate that the Opto-CRAC channel and its downstream effectors can be remotely activated using NIR light, thereby paving the way for its future applications in more (patho) physiologically-relevant mouse models, or ultimately, in cancer immunotherapies with improved spatiotemporal control over engineered therapeutic T cells or DCs to reduce the likelihood of off-tumor cross-reaction and to mitigate toxicity as well.

*5) The one area where UNCPs might have a real impact as NIR light transducers is for activating channelrhodopsin (in fact, the inability of ChR to be gated by long-wave light is given as an explicit motivation for UNCP development). Injected locally and recruited to genetically targeted cells that express channelrhodopsin, UNCP could enable NIR ChR gating. Surprisingly, this potentially exciting application is not explored.*

Because the main focus of this manuscript is on the Opto-CRAC system and its use in optogenetic immunomodulation, we hope that the editors and reviewers will agree with us that the use of UCNPs paired with ChR is beyond the scope of the current study. Nevertheless, we will extend a similar strategy to revamp ChR-based optogenetic tools in our near future work.

*6) Is the relatively slow time scale due to properties of ORAI channels, rather than a limitation of the light induction by STIM1-LOV? If so, the comparison to channelrhodopsin should probably be tempered. At least some discussion, and perhaps some experiments should be included as to which protein is limiting (both in terms of maximal conductance and kinetics).*

The relatively slow time scale is due to properties inherent to the ORAI1 channels (Hogan, Lewis and Rao, 2010; Prakriya and Lewis, 2015). Hence, as per the reviewers’ suggestion, we have tempered the comparison to channelrhodopsin in our revised manuscript. Still, we would emphasize that our system is a new and complementary addition to the existing optogenetic toolkits.

*7) STIM1 has other targets besides Orai1 channels including other Orai isoforms, voltage-activated Ca^2+^ channels (which it inhibits), and even TRP channels. How the optically responsive STIM1 (LOVSoc) fits into this larger framework of potential targets is unclear. Since other channels may be engaged by STIMs, the authors should examine calcium selectivity by testing for other ions in cells that have additional endogenous channels, as opposed to using fibroblasts stably expressing ORAIs.*

We measured the light-induced current by whole-cell recording in HEK293-ORAI1 cells (Figure 1). The I-V curve of LOVSoc-expressing cells exhibited a typical inward rectifying current (distinct from the large outward cationic currents of TRP channels), which is characteristic of the CRAC channel. In addition, replacing the most abundant extracellular ion, Na^+^, with a non-permeant ion NMDG^+^ did not significantly alter the amplitude and overall shape of the CRAC current, implying that Na^+^ has negligible contribution to LOVSoc-mediated photo-inducible CRAC current in our system. Voltage-gated Ca^2+^ channels are not a concern in our system, as they are barely expressed in cells of the immune system or most of other non-excitable tissues, and we did not detect any voltage activated Ca^2+^ currents during our whole-cell recording.

*8) The vector size appears small enough for viral gene delivery, but it is unclear how UCNPs can be delivered. In addition, if the UCNPs binds to off-targets, that could wreak havoc with the high-energy blue light in the body.*

The current application is, but not restricted to, ex vivo treatment and adoptive cell transfer back to the body for therapeutic or interventional purposes. The UCNPs loaded cells are consequently introduced by i.v. or subcutaneous injection. In the long term, the targeted delivery of UCNPs can be achieved by elegant nanoparticle surface modification with targeting moieties (Anal. Chem. 2009, 81, 8687; ACS Nano 2011, 5, 3744; or Chem. Commun. 2006, 28, 2557). Furthermore, owing to the spatial and temporal accuracy of NIR light activation, one can always use guided NIR light to confine localized blue light generation. In this way, the photoactivation of off-target regions can be minimized. Moreover, compared to channelrhodopsin, which routinely requires a power intensity of >0.5-1 mW/mm^[3]^ in order to obtain a steady-state response, our Opto-CRAC system only requires 0.04 mW/cm^[3]^ blue light to elicit calcium influx and downstream effects. Thus, the potential toxicity caused by blue light is minimized.

*9) Regarding the UNCPs: the conversion here is from low energy to high, which could be highly inefficient. Energies of excitation and emission at different wavelengths should be included.*

In accordance with the reviewers’ suggestion, we have quantified the upconversion quantum efficiency by following our previous method (ACS Nano 2014, 10621-10630). The upconversion quantum efficiency (QE) can be utilized to precisely measure the upconversion ability of the characterized materials. This is defined as a fraction of the absorbed photons that successfully employed to generate upconversion emission. A detailed description of QE calculation using our previous protocol was added to the Methods section. The upconversion QE of UCNPs used in the current study within the blue region was determined to be 2.7%.